# Polo-like kinase 1 coordinates biosynthesis during cell cycle progression by directly activating pentose phosphate pathway

Xiaoyu Ma[1], Lin Wang[1], De Huang[1], Yunyan Li[2], Dongdong Yang[1], Tingting Li[1], Fudong Li[3], Linchong Sun[1], Haoran Wei[1], Kun He[4], Fazhi Yu[1], Debiao Zhao[3], Lan Hu[1], Songge Xing[1], Zhaoji Liu[1], Kui Li[1], Jing Guo[1], Zhenye Yang[1], Xin Pan[4], Ailing Li[4], Yunyu Shi[3], Junfeng Wang[2], Ping Gao[1] & Huafeng Zhang[1]

Two hallmarks for cancer cells are the accelerated cell cycle progression as well as the altered metabolism, however, how these changes are coordinated to optimize the growth advantage for cancer cells are still poorly understood. Here we identify that Polo-like kinase 1 (Plk1), a key regulator for cell mitosis, plays a critical role for biosynthesis in cancer cells through activating pentose phosphate pathway (PPP). We find that Plk1 interacts with and directly phosphorylates glucose-6-phosphate dehydrogenase (G6PD). By activating G6PD through promoting the formation of its active dimer, Plk1 increases PPP flux and directs glucose to the synthesis of macromolecules. Importantly, we further demonstrate that Plk1-mediated activation of G6PD is critical for its role to promote cell cycle progression and cancer cell growth. Collectively, these findings establish a critical role for Plk1 in regulating biosynthesis in cancer cells, exemplifying how cell cycle progression and metabolic reprogramming are coordinated for cancer progression.

[1] Hefei National Laboratory for Physical Sciences at Microscale, CAS Key Laboratory of Innate Immunity and Chronic Disease, Innovation Center for Cell Signaling Network, School of Life Sciences, University of Science and Technology of China, Hefei 230027, China. [2] High Magnetic Field Laboratory, Chinese Academy of Sciences, 350 Shushanhu Road, Hefei, Anhui 230031, China. [3] Hefei National Laboratory for Physical Sciences at Microscale and School of Life Sciences, University of Science and Technology of China, Hefei 230027, China. [4] Institute of Basic Medical Sciences, National Center of Biomedical Analysis, Beijing, 100850, China. Xiaoyu Ma, Lin Wang, De Huang and Yunyan Li contributed equally to this work. Correspondence and requests for materials should be addressed to J.W. (email: junfeng@hmfl.ac.cn) or to P.G. (email: pgao2@ustc.edu.cn) or to H.Z. (email: hzhang22@ustc.edu.cn)

For cells to proliferate, they must cycle through G1, S, G2 phases, and then mitosis, to divide into two daughter progenies. Meanwhile, given the energy and biosynthesis required to replicate the entire cellular contents, metabolic activity is increasingly appreciated as a major determinant of a cell's "decision" to proliferate or exit the cell cycle[1–6]. In the past decades, tremendous evidence has accumulated for the understanding of the machinery behind the cell cycle control, in particular, a series of G1, S, or G2 phase-specific checkpoint proteins have been identified[7–10]. Recent evidence also suggests that crosstalk occurs between cell cycle and metabolic control[4–6,11–14], pointing to the existence of a complicated network of cell cycle signaling that is cross talked with metabolic inputs. Nevertheless, the mechanisms are still poorly understood. For a better understanding and control of the cell proliferation and cancer progression, we are yet to define more specific regulators that potentially drive tumorigenesis both through cell cycle control and metabolic regulation.

Polo-like kinase 1 (Plk1) is a critical regulator of cell cycle and is highly expressed in proliferating cells[15,16]. Increasing evidence suggests that Plk1 is also involved in other cellular events in addition to mitosis. For instance, Plk1 functions to regulate DNA replication[17,18] and glycolysis indirectly through its target protein PTEN[19] or other metabolic pathways[20]. Recently, we have deciphered several metabolic inputs underlying the altered biosynthesis and cell cycle progression in cancer cells[21–23]. Further search for regulators of biosynthesis during cell cycle progression led us to the identification of Plk1 as a master regulator of pentose phosphate pathway (PPP), a major biosynthesis pathway whose aberrant activation was described in various cancer cells[24–29]. We find that Plk1 directly phosphorylates glucose-6-phosphate dehydrogenase (G6PD) and promotes the formation of its active dimer, thereby increasing PPP flux, and NADPH and ribose production for the synthesis of macromolecules. Importantly, we further demonstrate that Plk1-mediated activation of G6PD is critical for its role to promote cell cycle progression and cancer cell growth both in vitro and in vivo, thus, elucidating a previously unappreciated mechanism by which Plk1 is connected to biosynthesis for cancer progression.

## Results

**Plk1 enhances PPP pathway and biosynthesis in cancer cells.** Although many molecules such as cyclin-CDK complexes have been identified to control cell proliferation[30], little is known regarding how biosynthesis is regulated to coordinate cell cycle progression in rapidly proliferating cells. Hence, we first set out to determine whether the activity of PPP, a major biosynthesis pathway that generates ribose 5-phosphate (R5P) for de novo synthesis of nucleotides and NADPH from glucose catabolism, varies at different phases of cell cycle. HeLa cells were synchronized with double hydroxyurea (HU) block (12-h treatment with HU, 10-h release, and a second HU block for 12 h) followed by releasing into G1/S boundary phase (0 h), S phase (5 h), and G2/M phase (10 h) (Fig. 1a, left panel). Consistent with previous reports[31,32], western blot using the lysates from synchronized cells revealed that Plk1 expression increased when cells entering into S phase and reached the highest level at G2/M phase (Fig. 1a, middle panel). G6PD, 6-phosphogluconolactonase (PGLS), and 6-phosphogluconate dehydrogenase (6PGD) catabolize the major steps in PPP, through which G6P is converted to ribulose 5-phosphate that reversibly isomerizes to R5P (Fig. 1a, right panel). Nevertheless, we detected no variations in the protein expression of G6PD, PGLS, and 6PGD during cell cycle progression (Fig. 1a, middle panel). Intriguingly, the enzyme activity of G6PD, the rate-limiting enzyme that catalyzes the conversion of glucose-6-

phosphate to 6-phosphate-gluconolactone, increased when cells were released into S phase (5 h after release) and reached maximal level at G2/M phase (10 h after release) (Fig. 1b, left panel). However, the enzyme activity of 6PGD was not changed with the cell cycle progression (Fig. 1b, right panel). Treatment with nocodazole, a specific prometaphase arrest inducer, also markedly elevated the cellular G6PD activity in HeLa cells (Supplementary Fig. 1a). Consistent with G6PD activity, further analysis revealed that cellular NADPH levels increased at S phase (5 h after release) and reached maximal level at G2/M phase (10 h after release), accompanied by reduction in $NADP^+$/NADPH ratios (Fig. 1c). Similar results were obtained in Hep3B cells (Supplementary Fig. 1b). These results suggested a correlation between PPP activation and the cell cycle progression.

Next, we sought to determine how G6PD/PPP activity was regulated during cell cycle progression. We observed that when Plk1 inhibitor BI2536 was added in HeLa cell culture medium after the double HU block synchronization, the increase in G6PD activity and NADPH levels along with the reduction in $NADP^+$/NADPH ratios was eliminated either at 5, 10, or 12 h after release, corresponding to S, G2/M phases, and normal cell cycle progression, respectively (Fig. 1d, e). However, interestingly, cell cycle analysis revealed that BI2536-treated cells were able to proceed through cell cycle to S (5 h) and G2/M (10 h) phases, respectively, but remained at G2/M phase while dimethylsulfoxide (DMSO)-treated cells proceeded further to regular cell cycle at 12 h after release (Fig. 1f), which was consistent with previous reports that Plk1-null cells were arrested at G2/M phase. The delayed cell cycle progression in BI2536-treated cells was also confirmed by immunofluorescence staining of Phospho-Histone H3 (Ser10) (pH3Ser10) and cyclin B1 expression, which are the markers of late G2 and mitosis (G2/M) (Supplementary Fig. 1c, d). These results indicated that the variations in G6PD activity and NADPH levels at different cell phases are Plk1-dependent. Indeed, when Plk1 was overexpressed, G6PD activity and cellular NADPH levels were gradually increased in a Flag-Plk1 dose-dependent manner (Fig. 1g, h), whereas Plk1 small hairpin RNAs (shRNAs) significantly decreased G6PD activity and cellular NADPH levels in HeLa cells (Fig. 1i, j). Similar results were observed in Hep3B cells (Supplementary Fig. 1e, f). Consistent with NADPH changes, Plk1 expression affected the levels of glutathione (GSH) and reactive oxygen species (ROS) as well (Supplementary Figs. 1g–i and 2a, b). Of note, the 6PGD enzyme activity was not affected by Plk1 overexpression or Plk1 knockdown by tet-inducible shRNAs, suggesting that 6PGD was not involved in Plk1-regulated PPP metabolism (Supplementary Fig. 2c, d). Thus, we next focused on G6PD for further study.

To confirm that PPP flux fluctuates during the cell cycle progression, we performed metabolic tracing of [U-$^{13}$C]-labeled glucose in HeLa cells synchronized at different phases of cell cycle by liquid chromatography-mass spectrometry (LC-MS). As a result, we found that M + 5-labeled fraction of ribose-5-phosphate (R5P) was increased when cells entering into S phase and reached the highest level at G2/M phases (Fig. 2a), suggesting that HeLa cells rapidly acquired the capacity of catabolizing [U-$^{13}$C]-labeled glucose into R5P, a critical metabolite of PPP, during cell cycle progression. Consequently, M + 5-labeled fraction of the downstream nucleotides generated from the PPP flux, such as AMP, dTMP, and dCMP were significantly increased in HeLa cells at G2/M phases (Supplementary Fig. 2e).

To provide direct evidence that Plk1 stimulates PPP pathway, we measured lactate produced from PPP pathway using the [2-$^{13}$C]-labeled glucose by nuclear magnetic resonance (NMR) spectroscopy, which could distinguish the lactate produced from PPP and that derived from the general glycolysis pathway

(Fig. 2b). As a result, the NMR metabolic measurement revealed that forced expression of Plk1 elevated the PPP-derived lactate (Fig. 2c), whereas knockdown of Plk1 by tet-inducible shRNAs decreased the lactate produced from PPP pathway (Fig. 2d). Further NMR metabolic measurement using [U-$^{13}$C]-labeled glucose showed that overexpression of Plk1 enhanced the U-$^{13}$C incorporation into R5P and the ribosyl moiety of adenine

nucleotides (AXP), uracil nucleotides (UXP), guanine nucleotides (GXP), and cytosine nucleotides (CXP) in HeLa cells (Fig. 2e). On the other hand, knockdown of Plk1 by tet-inducible shRNAs markedly reduced the U-$^{13}$C-incorporated R5P, as well as ribosyl moiety of AXP, UXP, GXP, and CXP in HeLa cells (Fig. 2f). Collectively, these data demonstrate that Plk1 enhanced PPP and biosynthesis in cancer cells. Nevertheless, it is known that R5P is

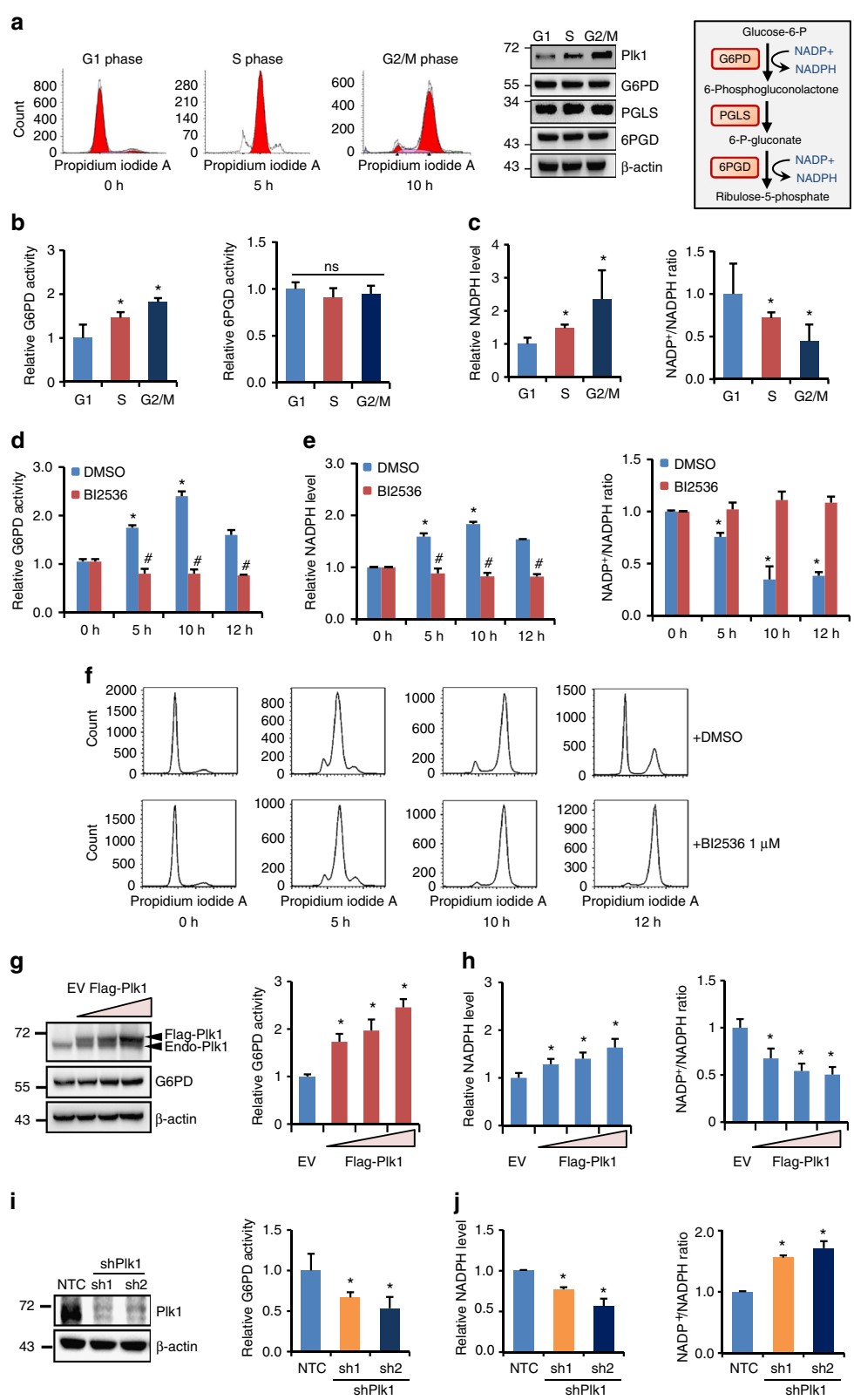

synthesized from glucose through both oxidative and non-oxidative PPP. Moreover, consistent with a previous report[19], we also observed that Plk1 expression stimulated cellular glycolysis in general (Supplementary Fig. 2f–k). To distinguish whether the changes in R5P is the consequence of G6PD-mediated oxidative PPP or a general enhanced glycolysis, we performed additional [13]C labeling metabolic flux assay using Plk1 overexpression and/ or G6PD knockdown cells. Our results showed clearly that Plk1 overexpression significantly enhanced R5P production, which was abolished by shG6PD (Fig. 2g). Consistently, shG6PD also significantly abolished the Plk1 overexpression-induced increase in the production of GXP, UXP, CXP, and AXP (Fig. 2g), suggesting that G6PD-mediated oxidative PPP was the major cause for changes in R5P and subsequent nucleotides levels regulated by Plk1. Taken together, our data demonstrate that Plk1, a critical cell cycle regulator, promotes oxidative PPP and biosynthesis in cancer cells.

**Plk1 activates G6PD by binding with and phosphorylating G6PD.** Next, we sought to determine the mechanisms by which Plk1 regulates G6PD activity and the PPP pathway. We observed that knockdown of G6PD by shRNAs obviously attenuated Plk1-induced increase in NADPH levels both in HeLa and Hep3B cells (Fig. 3a, b and Supplementary Fig. 3a, b), confirming that G6PD was involved in Plk1-regulated PPP metabolism. However, consistent with the western blot result in Fig. 1a, neither G6PD protein nor its mRNA expression was affected by Plk1 in HeLa cells (Supplementary Fig. 3c, d). As a kinase protein, Plk1 plays vital roles by phosphorylating downstream substrates. To study whether the regulation of Plk1 on G6PD activity is dependent on its kinase activity, the constitutively active Plk1 mutant (T210D) and the kinase-dead version of Plk1 (K82R) were employed (Supplementary Fig. 3e). Analysis of the PPP metabolic activity revealed that similar to wild-type (WT) Plk1, the constitutively active Plk1-T210D remarkably enhanced G6PD activity and NADPH levels, while the kinase-dead mutation of Plk1 (K82R) showed marginal effect on either G6PD activity or cellular NADPH levels (Fig. 3c, d), indicating that the regulation of G6PD activity and PPP metabolism by Plk1 was dependent on its kinase activity.

Co-immunoprecipitation experiments showed that Plk1 over-expressed in 293T cells specifically associated with G6PD protein (Fig. 3e). More importantly, our results also revealed that endogenous Plk1 interacted with G6PD in HeLa cells (Fig. 3f). In addition, pull-down assay using the purified recombinant proteins demonstrated the direct interaction between Plk1 and G6PD (Fig. 3g). As a kinase protein, Plk1 has a kinase domain (KD) at amino terminal and two polo-box domains (PBDs) at carboxyl terminal (Supplementary Fig. 3f). Using the vectors expressing the KD or PBD fragments of Plk1, we further demonstrated that G6PD bound to the PBD regions of Plk1

(Fig. 3h). Importantly, their interaction was enhanced when cells were treated with nocodazole, which arrested the cells to prometaphase (Supplementary Fig. 3g).

To identify the residues on G6PD protein responsible for its binding to Plk1-PBD, we first constructed vectors expressing WT, N-terminal (1–210), or C-terminal (211–515) of G6PD protein. Interestingly, similar to G6PD WT protein, both G6PD N-terminal and C-terminal were found to interact with Plk1-PBD domain (Fig. 3i). Plk1-PBD domain is known to recognize a consensus sequence of S-pS/pT-P/X[33–35] and analysis of G6PD protein revealed four potential PBD-binding sites (Ser180, Ser189, Thr279, and Thr333). To examine whether the four candidate sites of G6PD would mediate the interaction with Plk1-PBD, Ser180, Ser189, Thr279, and Thr333 were mutated to Ala individually and subjected to co-immunoprecipitation analysis. The results showed that G6PD T333A mutant displayed similar binding ability as WT G6PD with Plk1-PBD, while S180A, S189A, and T279A mutants abolished the interaction with Plk1-PBD (Fig. 3j). Thus, Ser180, Ser189, and Thr279 of G6PD seemed to be the key sites involved in its interaction with Plk1-PBD (Fig. 4a), which is consistent with our observation that Plk1-PBD interacted with both C-terminal and N-terminal of G6PD (Fig. 3i). To determine whether phospho-peptides on G6PD were required for its interaction with Plk1-PBD domain, we added PP2A phosphatase to cell extracts before performing immunoprecipitation in HeLa cells. The results showed that the binding ability of G6PD to Plk1-PBD was significantly reduced in the presence of PP2A phosphatase (Fig. 4b).

To further explore whether G6PD is a phosphorylation substrate of Plk1, we first conducted computational analysis using the group-based phosphorylation scoring method, which predicts that G6PD is a substrate of Plk1 and its T466, T406, and T327 are predicted as the potential phosphorylation sites by Plk1. On the other hand, mass spectrum analysis revealed that T406 and S296 of G6PD were potential phosphorylation sites by Plk1 (Supplementary Fig. 4a). Based on these analyses, we mutated four residues on G6PD protein, S296, T327, T406, and T466, to A or D, respectively, to determine the bona fide Plk1 phosphorylat-ing sites on G6PD experimentally. Forced expression of those mutants in HeLa cells revealed that similar to WT G6PD, both S296A/D and T327A/D increased the G6PD activity, however, T406A and T466A mutants failed to enhance the G6PD activity (Fig. 4c), suggesting that T406 and T466 are the potential Plk1 phosphorylating sites on G6PD. Consistently, T466D, imitating the active form, enhanced the G6PD activity (Fig. 4c). However, surprisingly, G6PD activity was significantly reduced in T406D mutant-expressing cells as compared with G6PD WT group (Fig. 4c). Similar results were observed in 293T cells (Supplementary Fig. 4b).

[32]P-phosphorylation assay using bacterially expressed G6PD protein and purified insect-expressed WT Plk1 protein and

**Fig. 1** Plk1 activates G6PD during cell cycle progression. **a–c** HeLa cells were synchronized into G1, S, and G2/M phases, and cells were harvested and subjected to western blot (**a**) and G6PD or 6PGD enzyme activity measurement (**b**), and NADPH and NADP$^+$/NADPH ratio levels (**c**) at indicated time after release. $n = 3$ biologically independent replicates. Data were presented as mean ± s.d. *$P < 0.05$ as compared to G1 phase group by two-sided Student's $t$-test. β-actin served as loading control. **d–f** HeLa cells were synchronized at G0/G1 phase with double HU block. The G6PD activity (**d**), and NADPH and NADP$^+$/NADPH ratio levels (**e**) were measured in the presence or absence of 1 μM Plk1 inhibitor BI2536 at indicate hours after release. FACS analysis of cell cycle progression was shown in **f**. $n = 3$ biologically independent replicates. Data were presented as mean ± s.d. *$P < 0.05$ as compared to 0 h group, #$P < 0.05$ as compared to DMSO group by two-sided Student's $t$-test. **g, h** HeLa cells were transfected with empty vector (EV) or gradual amount of Flag-Plk1 plasmids. Cells were harvested and subjected to western blot and G6PD activity measurement (**g**), and NADPH and NADP$^+$/NADPH ratio levels (**h**). $n = 3$ biologically independent replicates. Data were presented as mean ± s.d. *$P < 0.05$ as compared to EV group by two-sided Student's $t$-test. β-actin served as loading control. **i, j** Plk1 protein levels and G6PD enzyme activity (**i**), and NADPH levels and NADP$^+$/NADPH ratio (**j**) were measured in HeLa cells stably expressing non-targeting control (NTC) or shPlk1. $n = 3$ biologically independent replicates. Data were presented as mean ± s.d. *$P < 0.05$ as compared to NTC group by two-sided Student's $t$-test. β-actin served as loading control

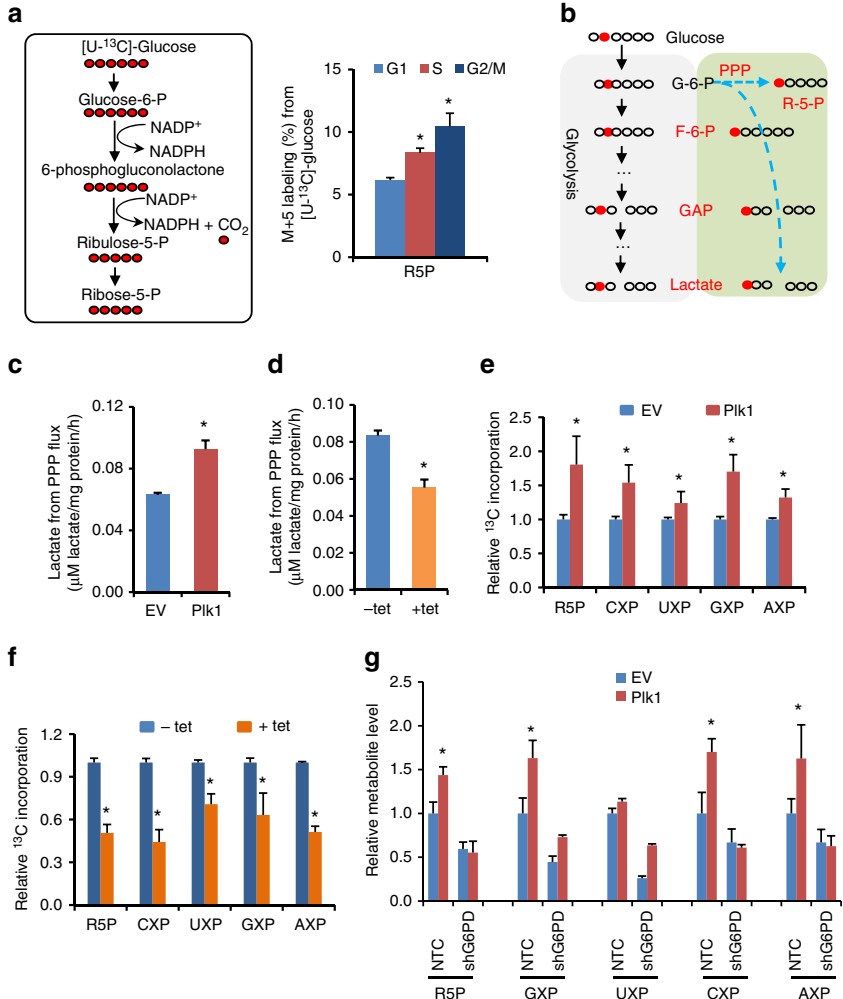

**Fig. 2** Plk1 enhances PPP and biosynthesis in cancer cells. **a** Using [U-$^{13}$C]-labeled glucose, M + 5-labeled R5P produced from PPP was detected by LC-MS. HeLa cells were synchronized by double HU block. When the cells were treated for the second round of HU block for 11 h, i.e., 1 h before releasing, U-$^{13}$C glucose was added into the medium and further culture for 1 h and then harvested for G1 phase cell analysis. For cell samples of S and G2/M phases, after the regular second round of HU block for 12 h, cells were released into fresh HU-free medium and further cultured for 4 and 9 h, respectively, followed by supplementation of U-$^{13}$C glucose and 1 h culture before harvesting cells for analysis. Cells were harvested and $^{13}$C-incorporated R5P was detected by LC-MS. The calculation of % of the m + 5 isotopologue of ribose is % of the sum of isotopologues. **b**–**f** The PPP metabolites were detected by NMR. Cells were cultured in medium containing [2-$^{13}$C]- or [U-$^{13}$C]-labeled glucose. The graphic description shows that by using the [2-$^{13}$C]-labeled glucose, NMR measurement could distinguish the lactate produced from PPP and that derived from the general glycolysis (**b**). Lactate derived from the oxidative PPP was calculated by the total lactate multiplied with the ratio of 3-$^{13}$C lactate detected by NMR in Plk1 overexpression (**c**) or knockdown cells (**d**). Using the [U-$^{13}$C]-labeled glucose, R5P and some downstream nucleotides generated from the PPP flux were detected by NMR in Plk1 overexpression (**e**) or knockdown cell lines (**f**). To induce Plk1 shRNA expression, cells were grown in the presence of 0.1 μg/ml doxycycline for 72 h. The calculation of isotopologue of ribose is total intensity of $^{13}$C-labeled ribose moiety. **g** U-$^{13}$C glucose metabolic flux assays were performed in Plk1-overexpressing HeLa cells with G6PD knockdown by NMR. The results were normalized to cell numbers or cell wet weight. Data were represented as the mean ± s.d. or s.e.m. *$P < 0.05$ as compared to indicated group by two-sided Student's $t$-test

kinase-dead mutation of Plk1 (K82R) demonstrated that WT G6PD protein was indeed phosphorylated by WT Plk1 in vitro judged by $^{32}$P incorporation, while kinase-dead mutation of Plk1 failed to increase $^{32}$P incorporation into WT G6PD protein (Fig. 4d). Moreover, compared with WT G6PD, the T406A and T466A mutants displayed less $^{32}$P incorporation in the presence of WT Plk1. Consistently, T406A/T466A double mutants displayed even less $^{32}$P incorporation, confirming that T406 and T466 are the Plk1 phosphorylating sites on G6PD (Fig. 4d). Moreover, we observed that G6PD threonine phosphorylation started to increase during S phase and reached the peak level at G2/M, which were markedly abolished by shPlk1 in HeLa cells, demonstrating that G6PD phosphorylation was affected by Plk1

in endogenous conditions (Fig. 4e). Our data also revealed that restoring G6PD WT and T466D mutant expression, but not that of T406A, T406D, or T466A, significantly recovered the G6PD activity and cellular NADPH levels in endogenous G6PD-knockdown HeLa cells (Fig. 4f–h). More importantly, knockdown of Plk1 by shRNAs remarkably attenuated WT G6PD-induced increase in G6PD activity and NADPH level, however, it had no effect on G6PD T466D-mediated increase in G6PD activity and NADPH levels (Fig. 4f–h). Taken together, these results demonstrated that Plk1 activated G6PD and subsequent PPP metabolism by phosphorylating G6PD at the sites of T406 and T466.

**Plk1-mediated G6PD phosphorylation promotes its dimerization.** The G6PD enzyme is in equilibrium of inactive monomer and active dimer[36,37]. Our disuccinimidyl suberate (DSS)

crosslinking analysis revealed that HeLa cells overexpressing Plk1 displayed a strong ability to form G6PD dimers and a corresponding decrease in G6PD monomers (Fig. 5a). Consistently,

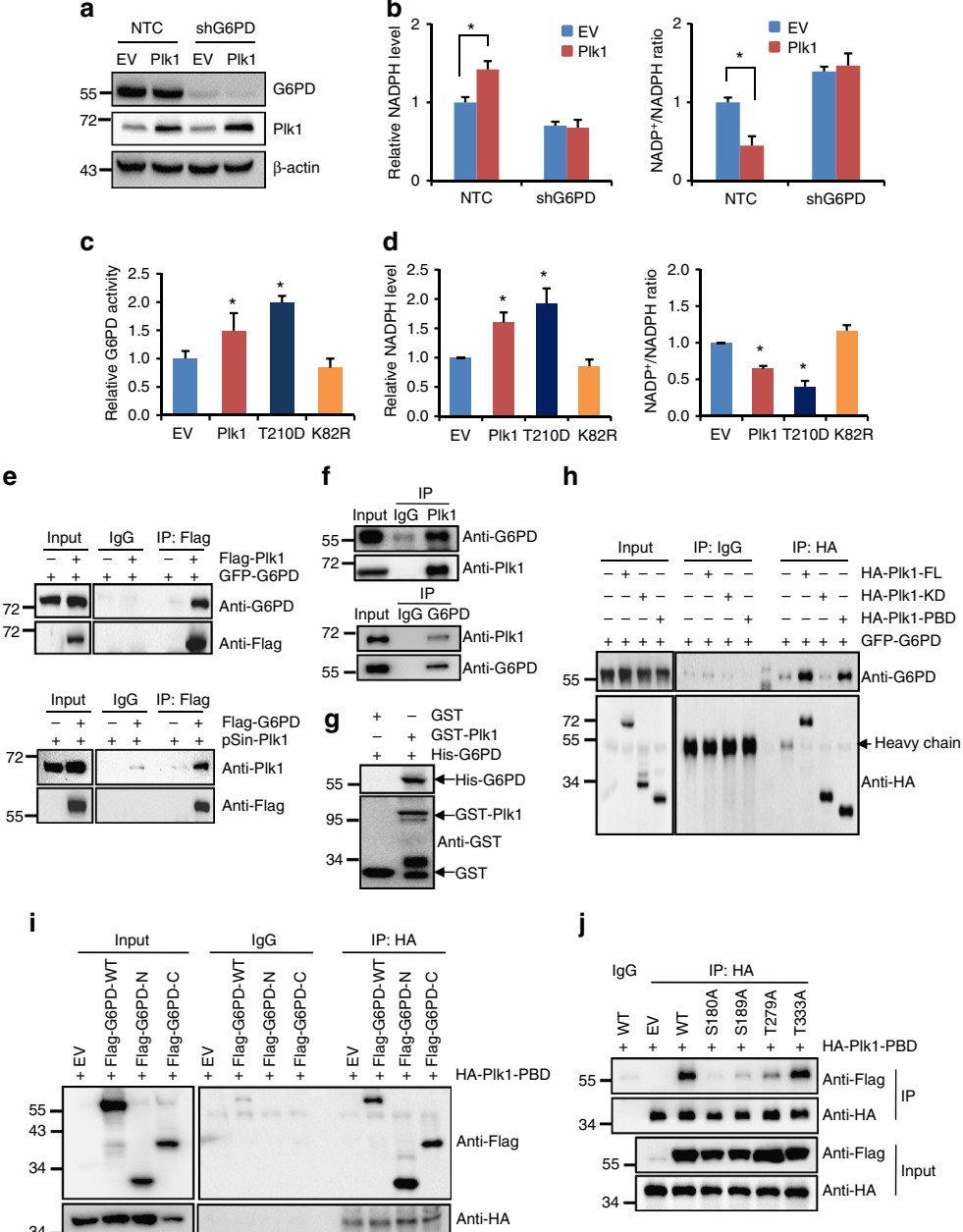

**Fig. 3** Plk1 regulates G6PD activity by interacting with G6PD. **a, b** HeLa cells stably overexpressing EV or Plk1 were further infected with viruses expressing NTC or shG6PD. Plk1 and G6PD protein (**a**), and relative NADPH and NADP$^+$/NADPH ratio levels (**b**) were determined. $n = 3$ biologically independent replicates. Data were presented as mean ± s.d. *$P < 0.05$ as compared between indicated groups by two-sided Student's $t$-test. **c, d** HeLa cells were stably overexpressing EV or Plk1 wild type or its T210D or K82R mutants. G6PD activity (**c**), and relative NADPH and NADP$^+$/NADPH ratio levels (**d**) were determined. $n = 3$ biologically independent replicates. Data were presented as mean ± s.d. *$P < 0.05$ as compared to EV group by two-sided Student's $t$-test. (**e**) 293T cells were transfected with eGFP-G6PD plasmids alone or together with Flag-Plk1 plasmids. Or 293T cells were transfected with pSin-Plk1 plasmids alone or together with Flag-G6PD plasmids. Cell lysates were immunoprecipitated with anti-Flag antibody or IgG, followed by western blot analysis. **f** HeLa cells were harvested and subjected to immunoprecipitation with anti-Plk1 or anti-G6PD, followed by western blot analysis with anti-Plk1 and anti-G6PD. **g** GST pull down of His-G6PD by GST-Plk1 using proteins purified in *E. coli* bacteria, followed by western blot analysis with anti-G6PD and anti-GST antibodies. **h** 293T cells were transfected with eGFP-G6PD plasmids alone or together with HA-tagged plasmids expressing Plk1 full length or its kinase domain or polo-box domain. Cell lysates were immunoprecipitated with anti-HA antibody or IgG, followed by western blot analysis. **i** 293T cells were transfected with vectors expressing HA-Plk1-PBD and Flag-G6PD wild type or its truncated mutants as indicated. Cells were then harvested and subjected to immunoprecipitation analysis with anti-HA antibody or IgG, followed by western blot analysis with anti-Flag and anti-HA antibody. **j** 293T cells were transfected with vectors expressing HA-Plk1-PBD and Flag-G6PD wild type or its point mutations as indicated. Cells were then harvested and subjected to immunoprecipitation analysis with anti-HA antibody or IgG, followed by western blot analysis with anti-Flag and anti-HA antibody

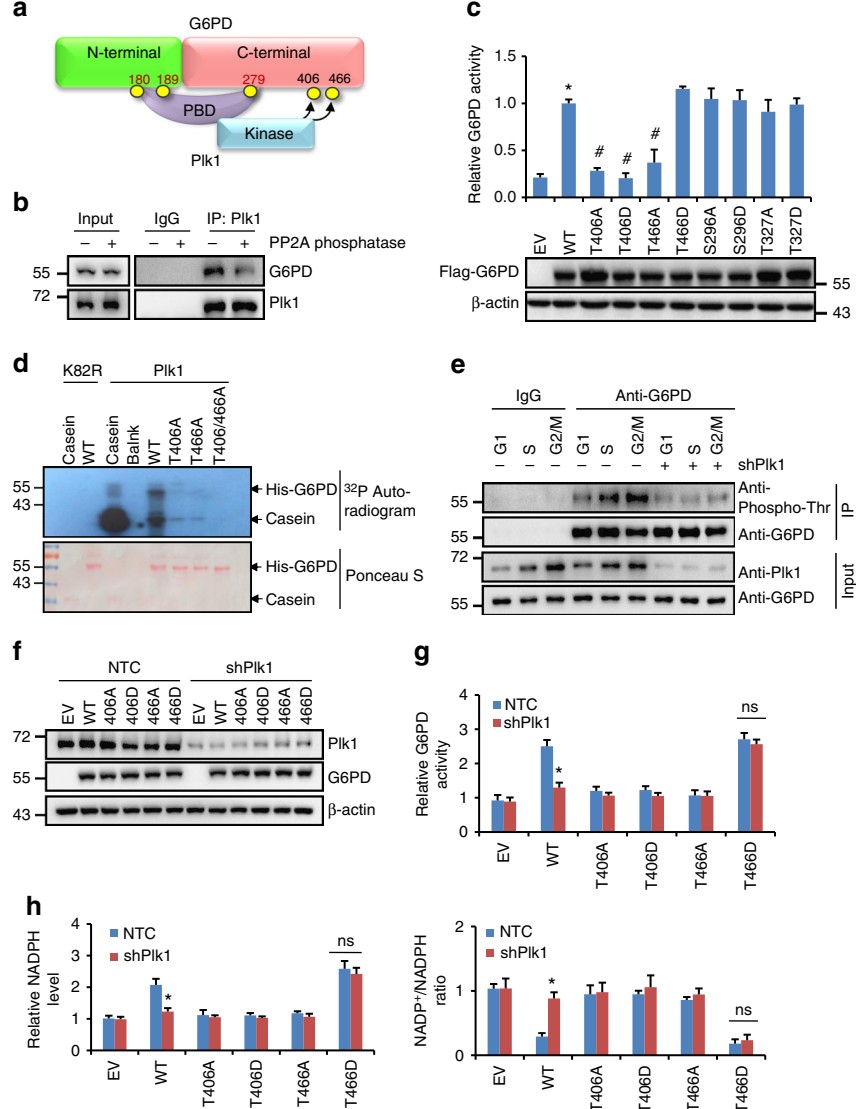

**Fig. 4** Plk1 regulates G6PD activity by phosphorylating G6PD. **a** The model depicting that the PBD domain of Plk1 protein binds to G6PD via the phosphorylation of its S180, S189, and T279 sites, and the kinase domain of Plk1 phosphorylates T406, T466 of G6PD. **b** HeLa cells were harvested and lysed. The cell lysates were separated into two parts, treated with or without PP2A phosphatase, followed by IP with anti-Plk1 and western blot analysis with G6PD and Plk1 antibodies. **c** HeLa cells expressing shG6PD were further infected with viruses expressing G6PD wild type or its mutants. The G6PD activity and protein expression were detected in those cells. Data were presented as mean ± s.d. *$P < 0.05$ as compared to EV group, #$P < 0.05$ as compared to WT group by two-sided Student's $t$-test. β-actin served as loading control. **d** Bacterial-expressed G6PD wild type and mutants were subjected to in vitro kinase assay using $^{32}$P labeling. Kinases used were insect-expressed wild-type Plk1 (active form) and K82R (inactive form). Casein acted as positive control, and the loading controls are shown in the bottom panel (stained by Ponceau S). **e** HeLa cells stably expressing NTC or tet-inducible shPlk1 were synchronized at G1, S, and G2/M phases. Cell were harvested and subjected to immunoprecipitation with anti-G6PD antibody, followed by western blot with G6PD and pan-phosphor-threonine antibody. Cells were grown in the presence of 0.1 μg/ml doxycycline to induce Plk1 shRNA expression. **f–h** HeLa cells stably expressing shG6PD were further transfected with vectors expressing G6PD wild type or its mutants together with NTC or shPlk1 as indicated. Cells were harvested and subjected to western blotting using Plk1 and G6PD antibodies (**f**), G6PD enzyme activity measurement (**g**), and NADPH level and NADP⁺/NADPH ratio measurements (**h**). $n = 3$ biologically independent replicates. Data were presented as mean ± s.d. *$P < 0.05$ as compared between indicated groups by two-sided Student's $t$-test

downregulation of Plk1 by tet-inducible shRNAs led to an obvious decrease in G6PD dimer formation and a strong increase in G6PD monomers (Fig. 5b), suggesting that Plk1 promotes the formation of G6PD dimers. Analysis of the interaction between two differentially tagged G6PD proteins, Flag-G6PD and GFP-G6PD, also demonstrated that overexpression of Plk1 enhanced the interaction of these two differentially tagged G6PD proteins (Fig. 5c), while Plk1 inhibition by BI2536 suppressed their interaction in a dose-dependent manner (Fig. 5d). Moreover, DSS crosslinking analysis using synchronized HeLa cells demonstrated

that the G6PD dimer formation increased gradually as the cells entering into S phase and reached the highest levels at G2/M phase, with a corresponding decrease in G6PD monomer levels (Fig. 5e). Similar results were obtained in Hep3B cells (Supplementary Fig. 5a). Meanwhile, BI2536 treatment significantly blocked the increase in endogenous G6PD dimer formation at S phase and G2/M phases in synchronized HeLa cells (Fig. 5e). Further, nocodazole treatment, which arrested cells at prometa-phase, also increased significantly the interaction between two differentially tagged G6PD proteins, Flag-G6PD and GFP-G6PD,

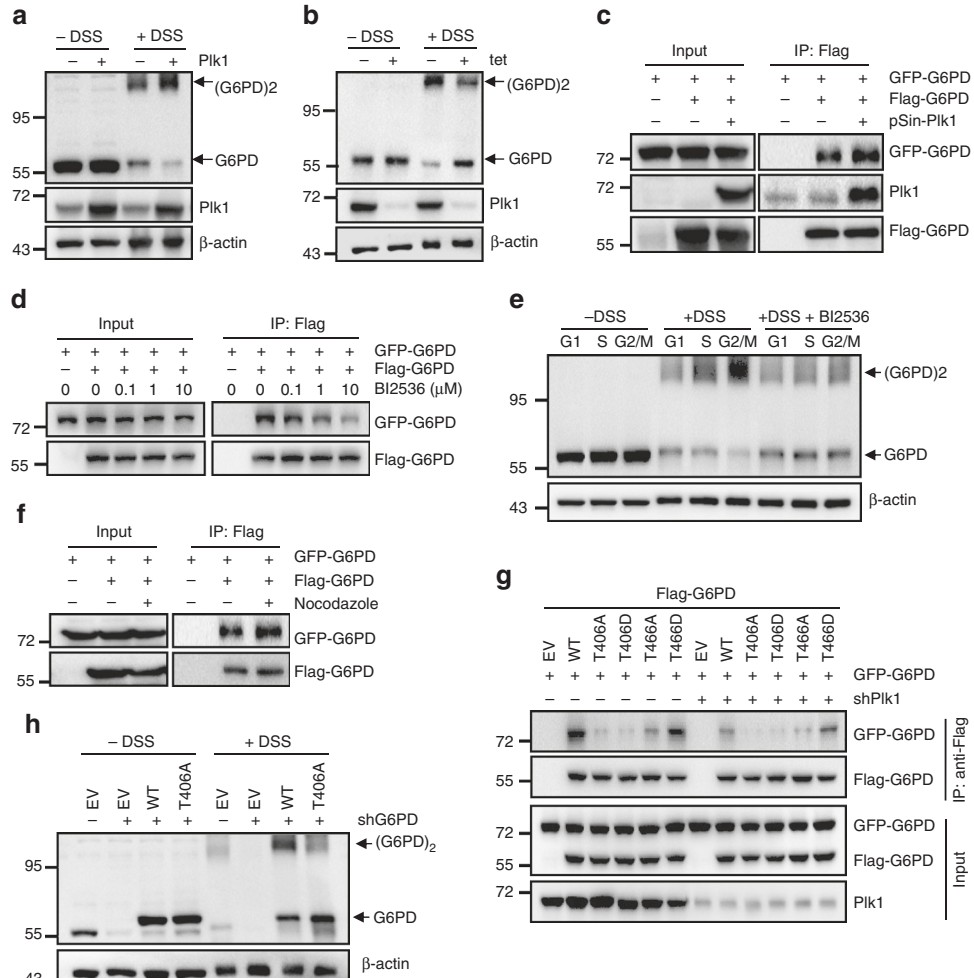

**Fig. 5** Plk1-mediated G6PD phosphorylation promotes its dimerization. **a**, **b** HeLa cells overexpressing Plk1 (**a**) or expressing tet-inducible shPlk1 (**b**) were treated with or without 1 mM disuccinimidyl suberate (DSS), followed by western blot analysis with antibodies against G6PD or Plk1. To induce Plk1 shRNA expression, cells were grown in the presence of 0.1 μg/ml doxycycline for 72 h. β-actin served as loading control. **c** HeLa cells transfected with eGFP-G6PD and Flag-G6PD vectors were co-transfected with pSin-Plk1 plasmid. Cell lysates were immunoprecipitated with anti-Flag antibody, followed by western blot analysis with antibodies against G6PD or Plk1. **d** GFP-G6PD and Flag-G6PD co-transfected HeLa cells were treated with BI2536 for 16 h at indicated concentrations. Cells were harvested and subjected to Co-IP using Flag antibody, followed by western blotting with Flag or GFP antibody. **e** HeLa cells were synchronized using HU double block during which vehicle or 1 μM BI2536 was added 1 h before releasing. Cells were harvested and crosslinked using DSS, followed by western blotting with anti-G6PD. **f** HeLa cells transfected with eGFP-G6PD and Flag-G6PD were treated with nocodazole. Cell lysates were immunoprecipitated with anti-Flag antibody, followed by western blot analysis with antibodies against G6PD. **g** NTC- or shPlk1-expressing 293T cells were co-transfected with GFP-G6PD and Flag-G6PD wild type or mutants as indicated. Cells were harvested and subjected to Co-IP using Flag antibody, followed by western blotting with Flag or GFP antibody. **h** G6PD wild type or its T406A mutant was forced overexpressed in HeLa cells stably expressing shG6PD. Cells were treated with or without 1 mM DSS, followed by western blot analysis with antibodies against G6PD. β-actin served as loading control. NTC denotes non-targeting control

in HeLa cells (Fig. 5f). Considering the fact that dimer is the active form of G6PD, these data are in agreement with the changes of G6PD enzyme activity during cell cycle progression (Fig. 1b) and prove that Plk1 facilitates the formation of G6PD dimers.

To confirm that G6PD phosphorylation was involved in Plk1-regulated G6PD dimer formation, vectors expressing T406 or T466 mutated G6PD proteins were employed. Immunoprecipitation assay revealed that substitution of T406A or T466A drastically disrupted the interaction between G6PD subunits, but not T466D, a mutant that imitates constitutive phosphorylation (Fig. 5g). Furthermore, knockdown of Plk1 drastically decreases WT G6PD-G6PD interaction, while it had marginal effect on T466D mutant interaction (Fig. 5g). It should be noted that similar to T406A and T466A, substitution of T406D also

suppressed the interaction between G6PD subunits (Fig. 5g), which is consistent with their effects on G6PD activity (Fig. 4c and Supplementary Fig. 4b). It has been reported that the change of the G6PD dimer interface affects the stability and integrity of this active enzyme[38,39]. Interestingly, structural prediction analysis of the G6PD dimer reveals that T406 lies at the dimer interface (Supplementary Fig. 5b), positioning T406 as a critical site for G6PD dimer formation, which might explain why both T406A and T406D mutations impaired the dimer formation and activity of G6PD. In addition, DSS crosslinking experiment further demonstrated that T406A mutant displayed impaired ability to form dimers as compared to WT G6PD (Fig. 5h). Taken together, these data prove that Plk1-mediated phosphorylation of G6PD promotes the formation of active G6PD dimers.

**G6PD activity is vital for Plk1-regulated cell proliferation**. Live Cell Imaging System analysis revealed that knocking down G6PD markedly extended the duration time of mitotic cells (Fig. 6a). Furthermore, restoring the expression of G6PD WT or T466D mutant, but not that of T406A, T406D, or T466A, remarkably reduced the duration time of mitotic cells in endogenous G6PD-knockdown HeLa cells (Fig. 6b). G6PD is a critical enzyme in cellular PPP metabolism, which generates R5P for de novo synthesis of nucleotides and produces the essential intracellular reductant NADPH for neutralizing cellular ROS. To examine whether G6PD-mediated PPP metabolism was involved in the cell cycle progression, nucleoside mix (Nuc) containing four ribonucleosides (A, G, U, and C) plus four deoxynucleosides (dA, dG, dT, and dC) and/or ROS scavenger N-acetyl-l-cysteine (NAC) were supplemented in culture medium. Intriguingly, supplementation of NAC and Nuc mixtures in medium reduced the duration time of mitotic cells expressing T406A, T406D, and T466A to that of the G6PD WT or T466D mutant, underlining the importance of macromolecular synthesis via Plk1-mediated G6PD phosphorylation and activation during the cell cycle progression of tumor cells (Fig. 6b). Consistently, cell growth analysis revealed that restoring the expression of G6PD WT or T466D mutant, but not that of T406A, T406D, or T466A, markedly enhanced the growth of G6PD knockdown HeLa cells (Fig. 6c). Supplementation of NAC and Nuc mixtures in medium promoted the growth of cells expressing T406A, T406D, and T466A to the similar levels of G6PD WT- or T466D mutant-expressing cells (Fig. 6c). Of note, supplementation of nucleosides alone promoted the growth of cells expressing T406A or T466A mutants to the similar levels of the G6PD WT-expressing cells, but NAC alone showed marginal effect on cell growth (Supplementary Fig. 6a). Moreover, addition of NAC and Nuc mixtures released the cells arrested at S and G2/M phases by shPlk1, leading to similar cell cycle distribution as the non-targeting control cells (Fig. 6d), suggesting that G6PD-mediated PPP metabolic flux is important for Plk1-regulated cell cycle progression. Consistently, supplementation of nucleosides plus NAC or nucleosides alone recovered shPlk1-inhibited cell growth in HeLa cells (Fig. 6e and Supplementary Fig. 6b). Further, knockdown of G6PD attenuated Plk1 overexpression-induced increase in cell growth, which was recovered partially by further supplementation of nucleosides plus NAC or nucleosides alone (Fig. 6f and Supplementary Fig. 6c). Taken together, these results suggest that the regulation of Plk1 on cell cycle progression and cell proliferation depends on G6PD-mediated PPP pathway, at least partially.

It should be noted that supplementation of nucleosides alone, but not NAC alone, recovered cell growth significantly, suggesting that PPP-produced R5P was more critical for cell growth than NADPH in the cellular systems we tested. Since R5P is essential for nucleotides and subsequent DNA synthesis, we next monitored the DNA synthesis by measuring the cellular EdU incorporation. As a result, restoring the expression of WT G6PD, but not its 406A and 466A mutants, recovered the shG6PD-induced decrease in cellular EdU incorporation in HeLa cells (Supplementary Fig. 6d). However, supplementation of nucleosides mix in medium raised the cellular EdU incorporation in cells expressing G6PD 406A and 466A mutants to that of the WT G6PD-expressing cells (Supplementary Fig. 6d), further underlining the importance of macromolecular synthesis via G6PD phosphorylation and activation for Plk1-mediated cell cycle progression in tumor cells.

**G6PD is critical for Plk1-mediated tumor proliferation in vivo**. To address whether G6PD-mediated PPP pathway is important

for Plk1-regulated tumor growth, we performed xenograft experiments using Hep3B and HeLa cells. As previously reported[15,40], stable overexpression of Plk1 increased tumor size and tumor mass in Hep3B cells compared with control empty vector group (Supplementary Fig. 7a, b). However, when G6PD was knocked down, Plk1-enhanced tumor growth of Hep3B xenografts was markedly retarded (Supplementary Fig. 7a, b), suggesting that G6PD was critical for Plk1-regulated tumor growth in mouse model. Western blotting using the tumor tissue lysates confirmed the overexpression of Plk1 and the knockdown of G6PD by shRNAs in Hep3B xenografts (Supplementary Fig. 7c). Additionally, we further examined the effect of Plk1 inhibitor BI2536 on G6PD in vivo using xenograft model. We observed that while overexpressing G6PD promoted tumor growth in vivo, BI2536 treatment dramatically suppressed it even when G6PD was overexpressed (Supplementary Fig. 7d). Using the tumor tissues, we further confirmed that BI2536 treatment suppressed G6PD activity in vivo while it exerted no effect on G6PD protein levels (Supplementary Fig. 7e, f). These data further demonstrated that Plk1 inhibition suppressed G6PD activity in vivo to retard tumor growth.

To further dissect the Plk1-G6PD regulation axis in tumor growth, we incorporated the T466D mutant and performed additional experiments to study the tumor cell growth both in vitro and in vivo. Analysis of cell growth by trypan blue staining revealed that restoring the expression of WT G6PD or T466D mutant, but not that of T406A, T406D, or T466A, significantly enhanced cell growth in endogenous G6PD-knockdown HeLa cells (Fig. 7a). More importantly, knockdown of Plk1 by shRNAs obviously attenuated WT G6PD-stimulated cell growth, however, it had no effect on T466D mutant-mediated cell growth (Fig. 7b, c). Further in vivo xenograft experiments revealed that stable overexpression of WT G6PD or T466D mutant, but not that of T406A, T406D, or T466A, in endogenous G6PD-knockdown HeLa cells promoted tumor growth (Fig. 7d–g). Doxycycline (Dox) treatment administered in the drinking water, which suppressed Plk1 expression, dramatically abolished WT G6PD-promoted tumor growth (Fig. 7d–g). Importantly, however, T466D mutant-enhanced tumor growth was not affected by Dox treatment (Fig. 7d–g). These results underline the importance of the phosphorylation of G6PD regulated by Plk1 in tumor development. Western blotting using the tumor tissue lysates confirmed the knockdown of Plk1 and the overexpression of G6PD WT and mutants in HeLa xenografts (Fig. 7h). Similar results were observed in xenograft experiments using Hep3B cells, in which restoring the expression of WT G6PD, but not its 406A and 466A mutants, in G6PD knocking down cells promoted tumor growth (Supplementary Fig. 7g–i). Taken together, these results demonstrate that G6PD and its phosphorylation-related activation are critical for Plk1-mediated tumor growth in vitro and in vivo.

## Discussion

While the deregulated energy metabolism is widely appreciated in proliferating cells like cancer cells, surprisingly, most cancer cells have only a modest increase in their consumption of ATP compared to their need for precursor molecules and reducing equivalents for biosynthesis[11]. As a result, cancer cells must enhance macromolecular biosynthesis and coordinate it precisely with cell cycle progression. However, the mechanisms are still poorly understood. In this regard, it is significant for us to demonstrate that Plk1, a key regulator that is essential for mitosis, facilitates biosynthesis in cancer cells through activating PPP. Although multiple functions and numerous targets have been documented previously for Plk1[18,31,41,42], here for the first time,

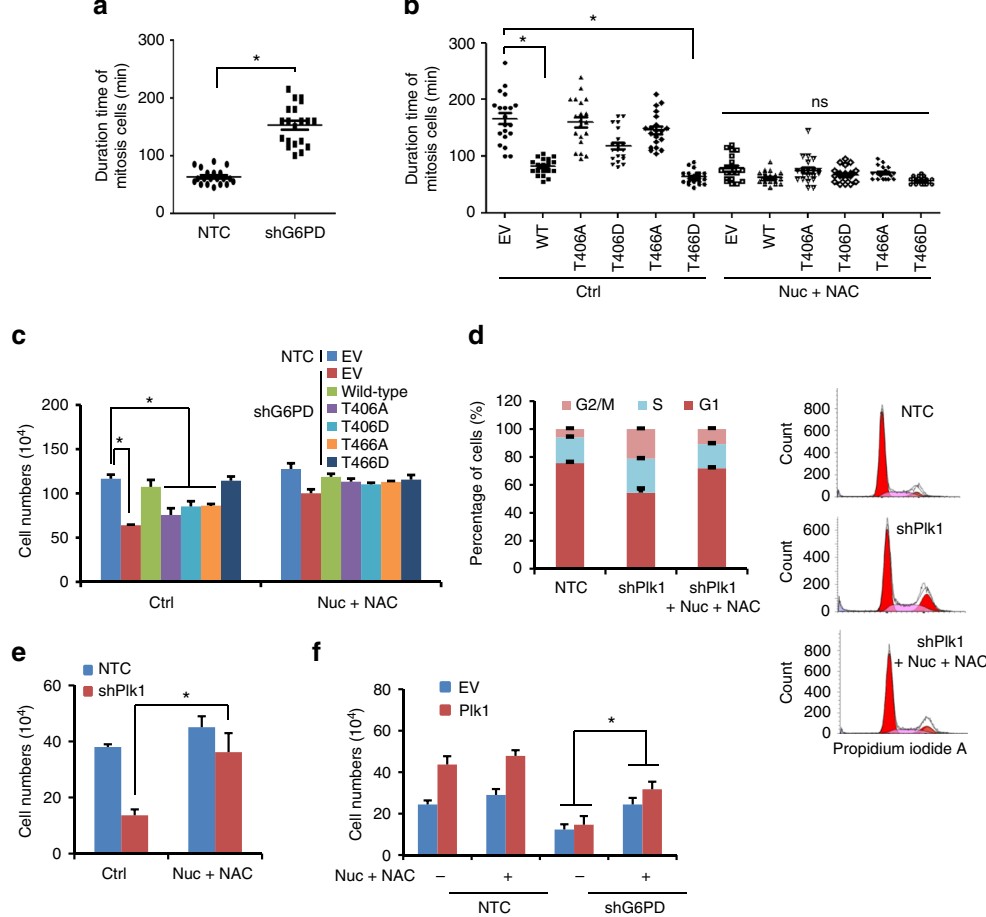

**Fig. 6** G6PD activity is vital for Plk1-regulated cell proliferation. **a** HeLa cells stably transfected with tet-inducible NTC or shG6PD (expressing RFP) were subjected to the living cell imaging station. To induce shRNA expression, cells were grown in the presence of 0.1 μg/ml doxycycline for 72 h. Then living cell imaging was done. The mitotic duration time of RFP positive cells was analyzed. n = 20 from 5 different fields. **b** HeLa cells stably expressing tet-inducible shG6PD (expressing RFP) were further transfected with vectors expressing eGFP-G6PD wild type and its mutants as indicated. Cells were treated with doxycycline in the presence or absence of NAC (2 mM) and Nuc mixtures (25 μM) in the medium, followed by living cell imaging. Images were recorded for 48 h. The mitotic duration time of GFP and RFP double-positive cells was statistically calculated. n = 20 from 5 different fields. **c** HeLa cells stably expressing NTC or shG6PD were further infected with viruses expressing empty vector (EV) or Flag-G6PD wild type or its mutants as indicated. Cells were treated with vehicle or Nuc mix (25 μM) and NAC (2 mM). Cell numbers were counted 4 days after treatment. Data were presented as mean ± s.d. *P < 0.05 as compared between indicated groups by two-sided Student's t-test. **d** HeLa cells stably expressing NTC or tet-inducible shPlk1 were grown in the presence of 0.1 μg/ml doxycycline to induce Plk1 shRNA expression. After 48 h of treatment, 2 mM NAC and 25 μM nucleosides mix were added together with doxycycline for another 24 h. Cell cycle distribution was monitored by flow cytometry. Representative histogram data and statistical results were shown. **e, f** HeLa cells stably expressing tet-inducible shPlk1 (**e**) or HeLa cells stably overexpressing Plk1 with further G6PD knockdown by shRNAs (**f**) were cultured in the medium supplemented with both 2 mM NAC and 25 μM nucleosides mix for 96 h. To induce Plk1 shRNA expression, cells were grown in the presence of 0.1 μg/ml doxycycline. Cell numbers were determined by trypan blue counting. Data were presented as mean ± s.d. *P < 0.05 as compared between indicated groups by two-sided Student's t-test

we provide sufficient evidence to link this critical kinase to biosynthesis for cell proliferation. First, we demonstrated that Plk1 increases PPP flux and biosynthesis from glucose catabolism; second, we established that G6PD is a direct target of Plk1 and proved that Plk1 regulates G6PD activity by interacting with and phosphorylating G6PD, which is required for the formation of active G6PD dimers; third, we found that G6PD activity is essential for Plk1-mediated cancer cell cycle progression in vitro and tumor growth in vivo. Collectively, our findings established the role of Plk1 in coordinating cell cycle progression with biosynthesis and thus substantially enriched the current understanding of functions of this kinase in the pathways to cancer development (Fig. 8).

It is interesting that we initially observed the fluctuation of G6PD activity during the cell cycle and that the activity of this enzyme is critical for cell cycle progression. Jiang et al.[43] also

reported recently that G6PD activity was critical for cell proliferation in p53-dependent manner. More recently, Du et al.[44] reported that p73, another member of p53 family, stimulated G6PD transcription that was associated with cell survival and proliferation. All these results placed G6PD at the crossroad for cell cycle progression. Of note, we confirmed repeatedly that consistent with Plk1 level, G6PD activity starts to increase during S phase, but reaches the peak level at G2/M instead of S phase, ensuring continuous cell cycle progression for dividing cells. It is likely that the biosynthesis process continues beyond S phase, as Atilla-Gokcumen et al.[45] documented recently that the lipid composition of cells in S phase differs drastically from cells in cytokinesis. It is well known that in continuously cycling cells, cell cycle is controlled tightly by cyclin-CDK complexes, which determine precisely the cell fate. To complete each cell cycle, the cells must also duplicate accurately the vast amount of DNA in

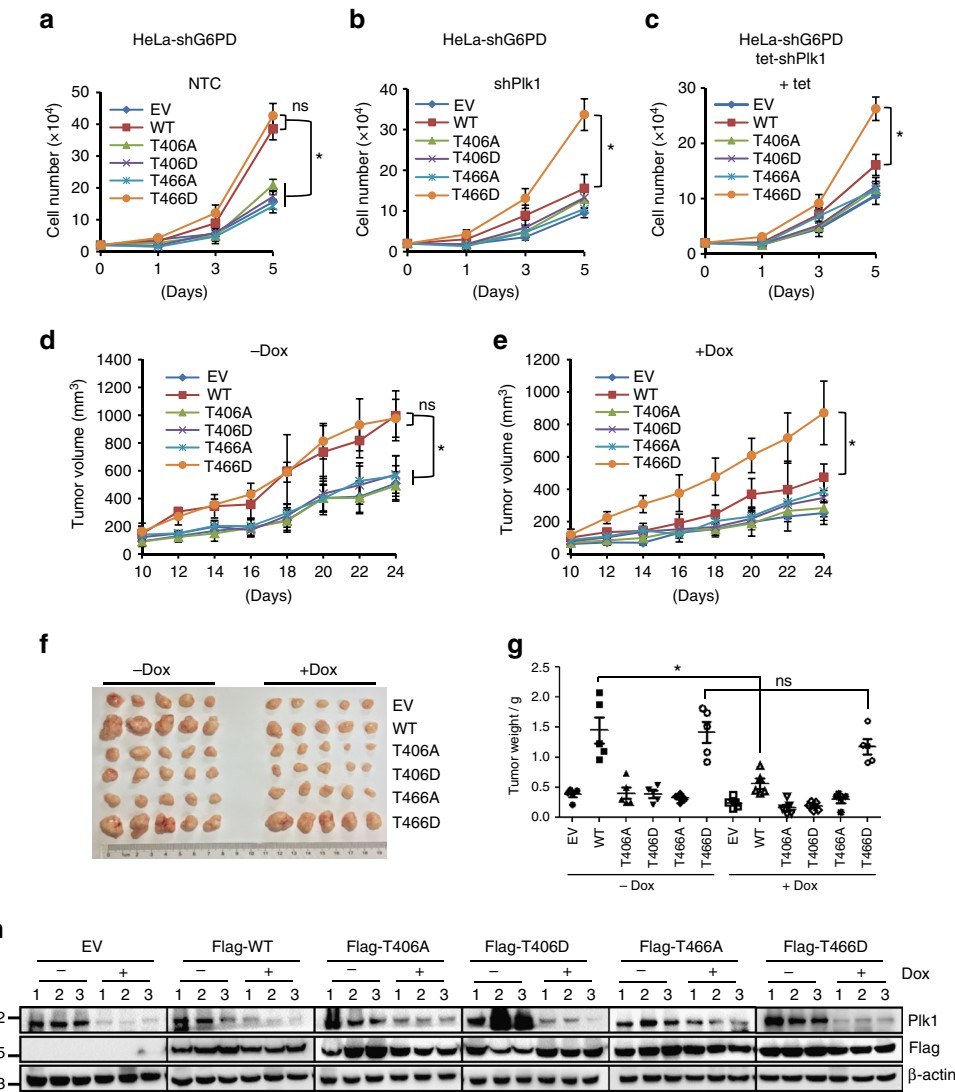

**Fig. 7** G6PD is critical for Plk1-mediated tumor proliferation in vivo. **a, b** HeLa cells stably expressing shG6PD were infected with viruses expressing pSin-3xFlag-G6PD wild type or mutants together with viruses expressing NTC (**a**) or shPlk1 (**b**) as indicated. Cell numbers were counted by trypan blue staining. $n = 3$ biologically independent replicates. Data were presented as mean ± s.d. *$P < 0.05$ as compared between indicated groups by two-sided Student's $t$-test. **c** HeLa cells stably expressing shG6PD were infected with viruses expressing pSin-3xFlag-G6PD wild type and mutants together with viruses expressing tet-inducible shPlk1 as indicated. Cells were treated with 0.1 μg/ml doxycycline and cell numbers were counted by trypan blue staining. $n = 3$ biologically independent replicates. Data were presented as mean ± s.d. *$P < 0.05$ as compared between indicated groups by two-sided Student's $t$-test. **d–g** HeLa cells stably expressing shG6PD were infected with viruses expressing pSin-3xFlag-G6PD wild type and mutants together with viruses expressing tet-inducible shPlk1 as indicated. Equal numbers of cells were subcutaneously injected into nude mice ($n = 5$ for each group). The mice were treated with or without doxycycline (1 mg/ml) in drinking water staring from 1 day before inoculation. The doxycycline-containing water was freshly prepared every other day. Tumor growth was measured starting from 10 days after inoculation (**d, e**). Tumors were extracted and compared at the end of the experiment (**f, g**). $n = 5$ for each group. Data were presented as mean ± s.d. *$P < 0.05$ as compared between indicated groups by two-sided Student's $t$-test. **h** Protein levels of Plk1 and G6PD were determined by western blot using the lysates of tumor tissues from each group as in **f** using anti-Plk1 and anti-Flag. β-actin served as loading control

the chromosomes and then segregate the copies precisely into two genetically identical daughter cells and, no doubt, any attempt to proliferate without sufficient macromolecular preparation would end in disaster for cells. Thus, it is very likely that there exist a series of metabolic or biosynthetic determinants that, in the similar way as cyclin-CDK complexes, control the cell cycle progression precisely. Intriguingly, emerging results have surfaced recently to show that several classical cell cycle related proteins such as Cyclin B1/Cdk1 or APC-Cdc20 have the function to modulate mitochondrial metabolic alterations[4,46,47]. Here, by defining the essential role of G6PD activation in Plk1-mediated cell cycle control and cancer progression, our results reinforced

the importance of crosstalk between cell mitosis and metabolic changes and, in particular, the significance of metabolic inputs in determining the cell fate. Obviously, we are just at the beginning to discover the metabolic checkpoints for cell cycle control and further insights of crosstalk is warranted to help fully understand the mechanism for cell proliferation and cancer development.

As macromolecular biosynthesis is an indispensable element for cell proliferation, it has long been the targets clinically for cancer therapy. Many clinically available chemical drugs such as 5-fluorouracil (5-FU) and methotrexate function as inhibitors of nucleotide synthesis. Unfortunately, most of these drugs are not specific to cancers, resulting in the toxic side effects to the non-

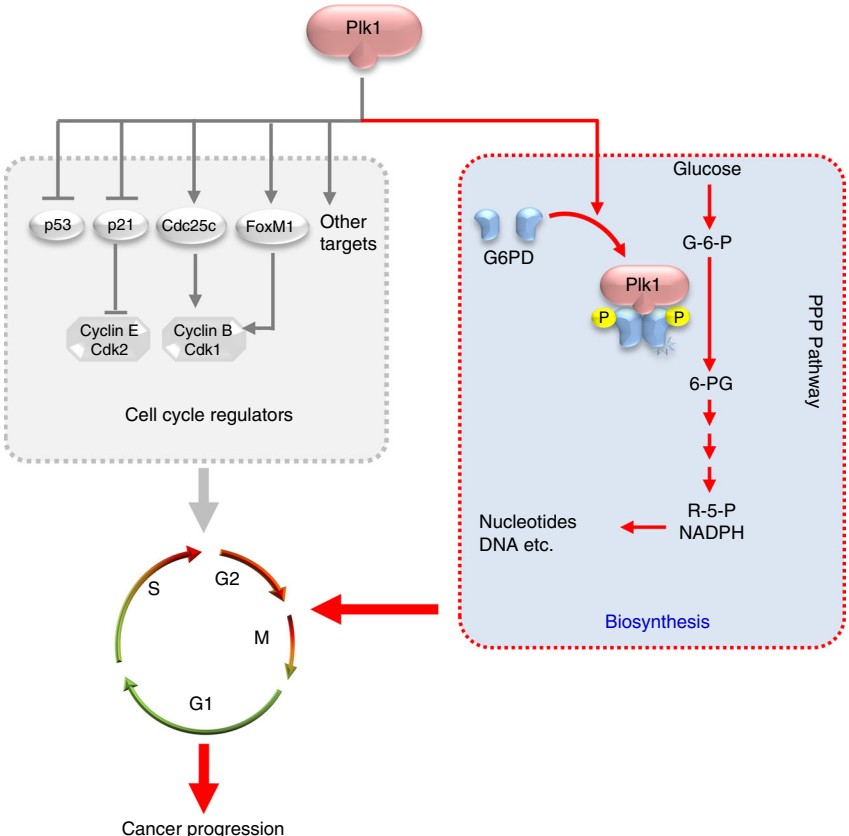

**Fig. 8** Working model. In this study, we demonstrate that Plk1 interacts with and directly phosphorylates G6PD to promote the formation of its active dimer, thereby enhancing PPP flux and macromolecule biosynthesis. Together with previous reports that have established multiple critical roles for Plk1 in regulation of mitosis and cell cycle progression, we propose that Plk1 coordinates biosynthesis with cell cycle progression to promote cancer development

cancer normal cells. Hence, it is very important to identify the mechanisms specifically underlying the deregulated macro-molecular biosynthesis in cancer cells. Plk1 expression and activity have been upregulated in numerous human malignancies[15,48–51]. With its established key role in facilitating cell cycle progression, Plk1 has been proposed as a target for cancer therapy. As a matter of fact, Plk1 inhibitors are on clinical trials with efficacies[52,53]. In an early report, Shao et al.[54] found that Plk1 inhibitor BI2536 treatment suppressed metformin-induced glycolysis and glutamine anaplerosis, both of which are survival responses of cells against mitochondrial poisons, impli-cating Plk1 inhibition to regulation of cancer metabolism. Here our novel results suggest that Plk1 inhibition could suppress cancer proliferation via both cell cycle progression and bio-synthesis. We thus envision that inhibiting Plk1 in combination with other chemicals that target metabolism and macromolecular biosynthesis in cancer cells could potentially provide unique rationale for cancer therapy.

## Methods

**Antibodies and reagents**. Antibodies and reagents used in the study are listed as following: anti-Plk1 (Sigma-Aldrich, P5998, 1:10,000), anti-Flag (Sigma, F3165, 1:5000), anti-HA (Sigma, H9658, 1:5000), anti-G6PD (Protein-tech, 25413-1-AP, 1:1000), anti-G6PD (Abcam, ab993, 1:1000), anti-glutathione S-transferase (anti-GST; Protein-tech, 10000-0-AP, 1:5000), anti-phospho-H3Ser10 (Millipore, 06-570, 1:1000), anti-cyclin B1 (Protein-tech, 55004-1-AP, 1:1000), anti-phospho-threonine (Millipore, AB1607, 1:1000), anti-PGLS (Santa Cruz, SC-398833, 1:1000), anti-6PGD (Protein-tech, 14718-1-AP, 1:1000), anti-CDK2 (Protein-tech, 10122-1-AP, 1:1000), rabbit IgG (Protein-tech), mouse IgG (Santa Cruz), and anti-Actin (Protein-tech, 60008-1-Ig, 1:5000). NADP⁺, NADPH, GSH, G6P, 6PG, NAC, Dox, tetracycline, HU, propidium iodide, nocodazole, 4,6-diamidino-2-pheny-lindole (DAPI), casein, $D_2O$, ribonucleosides, and deoxyribonucleosides were purchased from Sigma-Aldrich; R5P, nucleotides (AXP, GXP, CXP, and UXP, X =

M, D, and T) were purchased from Sangon Biotech. BI2536 was purchased from Selleck Corporation, PP2A was from Millipore, and human recombinant Plk1 was from Sino biological company. [2-$^{13}$C]-glucose and [U-$^{13}$C]-glucose were from Cambridge Isotope Laboratories. γ-$^{32}$P ATP was from China Isotope & Radiation Corporation.

**Plasmid construction**. Plk1, G6PD, and their mutants were cloned into pSin-EF2-puro vector for cell transfection and lentivirus production. pET22b vector and pGEX-4T1 were used for His-tagged and GST-tagged protein expression in bac-terial, respectively. ShRNA was cloned into plko.1 vector or tet-inducible pTRIPZ vector for gene knockdown study. For insect expression of Plk1 K82R, the CDS was cloned into pFastBac vector. For live-cell imaging, G6PD WT and its mutants were cloned into eGFP-C1 vector.

**Cell culture, plasmid transfection, and virus infection**. Hep3B, HeLa, and HEK293T cells were all obtained from American Type Culture Collection and maintained in Dulbecco's modified Eagle's medium (DMEM) supplemented with 10% fetal bovine serum (FBS, Bio-west), 1% penicillin, and streptomycin. All cells were cultured in a humidified incubator at 37 °C and 5% CO₂. All cell lines were tested for mycoplasma contamination and no cell lines were contaminated. To induce Plk1 and G6PD shRNA expression, cells were grown in the presence of 0.1 μg/ml Dox (Sigma). Lenti-viral plasmids pSin-Plk1 and pSin-G6PD (WT and mutants), were co-transfected with plasmids encoding group antigen, polymerase, envelope protein, and vesicular stomatitis virus G protein into HEK293T cells. Viral supernatant was collected 48 h post transfection, filtered (0.22 nm pore size), and added to Hep3B or HeLa cells in the presence of 8 μg/ml polybrene (Sigma-Aldrich). The transduced cells were selected by 0.6 μg/ml puromycin. ShRNAs targeting G6PD and Plk1 were commercially purchased (Sigma-Aldrich).

**Western blot**. Cells were harvested and total cellular protein was isolated using RIPA buffer (50 mM Tris-HCl, pH 8.0, 150 mM NaCl, 5 mM EDTA, 0.1% SDS, and 1% NP-40) supplemented with protease inhibitor cocktails (Roche). Protein concentration was measured using the Bradford assay kit. Equal amount of pro-teins were loaded and separated by SDS-polyacrylamide gel electrophoresis (SDS-PAGE). β-actin served as loading control. Uncropped images of immunoblots presented in this paper are provided in Supplementary Fig. 8.

**Quantitative real-time PCR**. Cellular total mRNA was extracted by Trizol (Invitrogen) and cDNA synthesis was performed using iScript cDNA synthesis kit (Bio-Rad). cDNA samples were used for quantitative real-time reverse transcription PCR (qRT-qPCR) analysis with iQSYBR Green Supermix and the iCycler Real-time PCR Detection System (Bio-Rad). For each primer pair, annealing temperature was optimized by gradient PCR. The fold change of target mRNA expression was calculated based on threshold cycle (Ct), where $\Delta Ct = Ct_{target} - Ct_{18S}$ and $\Delta(\Delta Ct) = \Delta Ct_{control} - \Delta Ct_{indicated\ condition}$. The relative mRNA levels were normalized to 18S. All primers were purchased from Invitrogen. The following primers were used in qRT-qPCR analysis for G6PD: forward: 5′-ACCGCA TCGACCACTACCT-3′; reverse: 5′-TGGGGCCGAAGATCCTGTT-3′.

**Cell synchronization and flow cytometry**. HeLa cells were synchronized using a HU double-block method. Briefly, the exponentially growing HeLa cells were maintained with 2 mM HU (Sigma-Aldrich) for 12 h, followed by a release of 10 h in fresh HU-free medium, and then cells were re-cultured in 2 mM HU for additional 12 h. Highly synchronized cells at the G1/S boundary were obtained. Then after a release of 5 h in fresh HU-free medium, the cell population entered into S phase. The G2/M phase cells were enriched at 10 h post release. Hep3B cells were synchronized using the following method: for G1 phase, cells were culture in serum-free DMEM for 48 h. For S phase, cells were culture in serum-free DMEM supplemented with 0.5 mM HU for 12 h, then release to complete DMEM containing 10% FBS for 6 h. For G2/M, cells were cultured in serum-free DMEM with nocodazole for 24 h. For cell cycle analysis, cells were trypsinized and pooled with the floating cells. Cells were washed with cold phosphate-buffered saline (PBS), fixed by ice-cold 70% ethanol, and suspended in a staining solution, which contains 20 μg/ml of propidium iodide, 0.1% Triton X-100, and 200 μg/ml of RNaseA, followed by analysis using a FACScan (BD Biosciences) and Modfit 2.0.

**Immunoprecipitation**. HeLa cells or 293T cells were lysed in buffer containing 20 mM HEPES (pH 7.5), 150 mM NaCl, 2 mM EDTA, 1.5 mM MgCl₂, 1% NP40, and protease inhibitors for 1 h at 4 °C followed by centrifugation. The supernatants were then diluted in a buffer without NP40 and pre-cleared by proteinA/G-Sepharose beads for 30 min. The supernatants were then incubated with indicated antibody for 4–6 h at 4 °C, followed by incubation with proteinA/G-Sepharose beads for 1 h at 4 °C. After incubation, beads were washed twice with lysis buffer, followed by further washing with ice-cold PBS, and boiling in 2× loading buffer. Protein samples were resolved by SDS-PAGE.

**EdU incorporation assay**. EdU incorporation assay was determined by using a Cell-Light EdU Apollo 488 In Vitro Imaging Kit (Ruibo Company, Shanghai, China) according to the manufacturer's instruction. The images were acquired with an Olympus ×51 microscope (Olympus), and EdU-positive cells were counted.

**Immunofluorescence**. Cells cultured on coverslips were pre-seeded 1 day before immunofluorescence analysis with the final confluence of 70–80%. Cells were fixed in 4% paraformaldehyde for 10 min, permeabilized with 0.1% Triton X-100 for 5 min, blocked with 5% bovine serum albumin and incubated with antibodies as indicated, followed by a Texas-red-conjugated anti-rabbit IgG and a fluorescein isothiocyanate-conjugated anti-mouse IgG antibody. The cells were mounted with DAPI-containing medium (Vector Laboratories) and the images were acquired with a microscope (Olympus).

**Live-cell imaging**. To study the mitotic duration time of HeLa cells, first of all, we established cell lines expressing red fluorescent protein (RFP; indicating the knockdown of endogenous protein). A unit of 0.1 μg/ml Dox was added for 72 h to induce pTripz Dox-inducible shG6PD (indicated by RFP). For the G6PD WT or mutant overexpression experiment, after 48 h of Dox treatment, cells were further transfected with vectors expressing eGFP-G6PD WT or mutants by Lipofectamine 2000 (Invitrogen), followed by incubation at 37 °C for 24 h. Then, cells were replated on glass coverslip. For live-cell imaging, coverslip-cultured cells were mounted in Rose chambers and maintained at 37 °C in phenol-free L-15 medium (Invitrogen) with 10% FBS and 0.1 μg/ml Dox. NAC and Nuc mixtures were supplemented before image collection. Time-lapse images were acquired at 5-min intervals for 48 h with a ×20 (0.50 numerical aperture) Plan-Fluor objective mounted on an Eclipse Ti microscope (Nikon). Mitotic entry was judged as the point of cell rounding, and mitotic exit was judged as the point at which cells flattened out. The mitotic duration time of RFP-positive cells (Fig. 6a) or RFP and green fluorescent protein (GFP) double-positive cells (Fig. 6b) was statistically calculated. Mitotic duration time of 20 cells from 5 different fields were shown.

**Crosslinking assay**. DSS crosslinking was used to detect the G6PD dimer. Briefly, cells were trypsinized and counted. Equal numbers of cells were collected for experiment. Cells were washed with cold PBS, followed by suspension in conjugation buffer (20 mM HEPES, pH = 8.0). The DSS solution prepared in DMSO was added to the cell suspension to a final concentration of 1 mM. After incubating at 37 °C for 30 min, the samples were boiled and used for western blot assay.

**G6PD enzyme activity measurement**. G6PD enzyme activity was determined by using a Biovision kit according to the manufacturer's instruction. Enzyme activities were normalized on the basis of protein concentration, which was determined by Bradford.

**6PGD enzyme activity measurement**. 6PGD activity was measured by the reaction converting NADP⁺ to NADPH in the presence of 6-phosphogluconate (6PG). The reaction buffer contained 50 mM Tris (pH 8.1) and 1 mM MgCl₂, and the substrates were 6PG (200 μM) and NADP⁺ (100 μM). Enzyme activities were normalized on the basis of protein concentration, which was determined by a Sangon Bradford protein assay kit.

**Metabolic measurement**. The extracellular lactate was measured using the cell culture medium with lactate assay kit (Biovision) according to the manufacturer's instruction. The values were normalized to the protein concentration. Intracellular NADPH and NADP⁺/NADPH ratio were measured using cell lysates with NADPH assay kit (Biovision) according to the manufacturer's instruction. The values were normalized to the protein concentration.

**LC-MS analysis of cell metabolites**. Roughly, $5 \times 10^6$ cells were washed twice with cold PBS, and polar metabolites were extracted by ice-cold 80% methanol immediately. Samples were treated with freeze and thaw cycle or sonication to sufficiently extract metabolites. The supernatant was collected and dried. The powder containing metabolites was dissolved in 80% methanol to run LC-MS. For the kinetic LC-MS analyses, a Shimadzu Nexera ×2 UHPLC combined with a Sciex 5600 Triple Time of Flight-Mass Spectrometry (TOFMS) was used, which was controlled by Sciex Analyst 1.7.1 instrument acquiring software. A Supelco Ascentis Express HILIC Acquity UPLC BEH Amide (150 cm × 2.1 mm, 1.7 μm) column was used with mobile phase (A) consisting of 5 mM ammonium formate and 0.05% formic acid; mobile phase (B) consisting of 90% acetonitrile (ACN) and 10% water. Gradient program: mobile phase (A) was held at 15% for 0 min and then increased to 30% in 7 min; then to 60% in 6 min and held for 1 min before returning initial condition. The column was held at 40 °C and 5 μl of sample was injected into the LC-MS with a flow rate of 0.2 ml/min. Automatic calibrations of TOFMS were achieved with average mass accuracy of < 2 ppm. Data Processing Software included Sciex PeakView 2.2, MasterView 1.1, and MultiQuant 3.0.2 (Supplementary Table 1). The calculation of % of the m + 5 isotopologue of ribose is % of the sum of isotopologues and the results were normalized to cell numbers.

**NMR analysis of ¹³C-labeled metabolites**. Cells were washed twice with cold PBS buffer. After centrifugation, the cell pellet was weighted (cell numbers are estimated to be roughly $3 \times 10^7$ per 100 mg wet weight, for most samples, 80–100 mg of wet weight cells were used) and resuspended in ice-cold 80% methanol immediately. Samples were treated with freeze and thaw cycles or sonication to sufficiently extract metabolites. The supernatant that containing polar metabolites was collected and lyophilized, and then dissolved in 50 μL PBS buffer (136 mM K₂HPO₄ and NaH₂PO₄, molar ratio of 4:1, pH 7.4 in D₂O) containing 0.01% TSP (m/v, sodium 3-trimethlysilyl [2,2,3,3-2H₄] propionate) for further NMR measurements. Here D₂O was used as a field lock, and TSP as an internal chemical shift reference.

All NMR experiments were performed on a Bruker AVANCE III 500 MHz NMR spectrometer at Chinese High Magnetic Field Laboratory (CHMFL, Hefei, China), equipped with 1.7 mm cryogenic CPTCI probe. All data were obtained at 298 K. Two-dimensional (2D) ¹H-¹³C heteronuclear single-quantum correlation (HSQC) spectra were recorded under identical acquisition parameters. The relaxation delay was 1.5 s, 128 scans were collected with 32 dummy scans. The 90° pulse width was about 13.2 μs for ¹H and 11.6 μs for ¹³C. The ¹H-¹³C J-coupling constant was set as 145 Hz. ¹H spectral width was 6 kHz (12 p.p.m.) whilst ¹³C was 170 p.p.m. GARP decoupling was applied during proton acquisition, and the 90° decoupling pulse width was 62.5 μs. An exponential line-broadening factor of 0.3 Hz was applied to all FIDs in t2, and zero filled to 2k prior to Fourier transformation. In t1, all data were zero filled to 2k, a Gaussian function of 0.5 Hz broadening factor together with an exponential function of 1 Hz were applied during t1 processing. The proton and carbon chemical shifts of internal reference TSP were set to 0 p.p.m. as the respective resonance reference.

To identify the resonances of individual metabolite, 2D HSQC experiments of 10 mM metabolite samples with standard compounds (such as G6P, ATP, R5P, et al) were collected and assigned. Standard chemical compounds were also added into cell extracts and increments of respective signal intensities further confirm the metabolite assignments (Supplementary Tables 2, 3). All assignments were verified by NMR experiments including 2D TOCSY, HMBC, 2D J-resolved ¹H spectroscopy and one-dimensional ¹H noesypr1d. To compare the relative concentration changes of interested metabolites, the peak volumes of cross-peaks in 2D HSQC spectra were integrated and normalized by cells wet weight.

**Protein purification and kinase assay and ³²P assay**. Protein expression was induced in BL21 bacteria by 0.5 mM isopropyl β-D-1-thiogalactopyranoside treatment at 16 °C. His-tagged proteins were purified using NTA Nickel column (Qiagen) and GST-tagged proteins were purified by GSH-conjugated agarose beads (GE). The purified proteins were concentrated. For kinase assay, the reaction was

performed in 20 μl of 1× kinase buffer (50 mM NaCl, 2 mM EGTA, 25 mM HEPES, pH 7.2, 5 mM MgSO$_4$, and 1 mM DTT) containing 100 ng of human recombinant Plk1, 2 μg of his-tagged protein and 50 μM ATP. For $^{32}$P assay, additional 5 μCi γ-$^{32}$P ATP was added into the kinase reaction mixture, followed by incubation for 30 min at 30 °C. The reactions were stopped with 5× loading buffer and separated by 10% SDS-PAGE. For LC-MS-MS, the gel was stained with Coomassie Bright Blue, and the proteins in the gel were cut out for phosphorylation sites analysis by LC-MS-MS. For $^{32}$P assay, the protein in gel was transferred to the nitro-cellulose membrane, followed by exposure to X-ray film.

**Nano LC-MS/MS**. Desalting and MS identification of digested peptides was carried out as described previously[4]. In brief, desalted peptides were analyzed using a TripleTOF 5600+ mass spectrometer (AB Sciex, Concord, Canada) coupled to an Ekspert TM nanoLC 425 (Eksigent, Dublin, OH). Five microliters of each sample was loaded onto a trap column (ChromXp C18, 350 μm × 0.5 mm, 5 μm, 120 Å, Eksigent, Dublin, USA) at a flow rate of 3 μl/min for 7 min. Peptide separation was carried out on a C18 column (ChromXp C18, 15 cm × 75 μm × 3 μm, 120 Å, Eksigent) at a flow rate of 0.3 μl/min. Peptides were separated using a 60 min linear gradient from 5% A to 35% B (mobile phase A, 1% formic acid; mobile phase B, 1% formic acid in ACN). The mass spectrometer was operated in positive ionization and high-sensitivity mode. The MS survey spectrum was accumulated from a mass range of 400–1250 $m/z$ in 250 ms. For information-dependent acquisition (IDA) MS/MS experiments, the first 20 features above 150 counts threshold and having a charge state of +2 to +5 were fragmented using rolling collision energy ±5%, with 100 ms spectra accumulation/experiment. Each MS/MS experiment put the precursor $m/z$ on a 15 s dynamic exclusion list. Autocalibration was performed every two samples (2 h) to assure high mass accuracy in both MS and MS/MS acquisition.

**Protein identification**. TripleTOF 5600 + raw data were processed using MS Data Converter (ABSciex) to generate.mgf files. All Mascot searches were performed against a downloaded SwissProt database (released in January 2015, 547,357 proteins, 194,874,700 residues) with taxonomy of *Homo sapiens*, using the following settings: trypsin with up to two missed cleavages; carbamidomethylation fixed at Cys; variable oxidation at Met; variable protein N-terminal acetylation, variable deamidation at Asn and Gln; variable phosphorylation at Ser, Thr, and Tyr. The mass error tolerance for precursor and fragment ions was set to 20 p.p.m. and 0.2 Da. A decoy reverse database search was also performed to estimate the false positive rate of protein identification. For unambiguous identification of proteins, the false discovery rate was set to 1%, and a matching of at least two peptides, each with more than 28 ion score.

**Animal experiments**. All animal studies were conducted with approval from the Animal Research Ethics Committee of the University of Science and Technology of China. Male or female BALB/c nude mice were purchased from SJA Laboratory Animal Company of China, which were randomly assigned to experimental groups. For xenograft experiments, Hep3B or HeLa cell lines were infected with indicated viruses expressing interested proteins or shRNAs. Equal numbers of the established stable cells were injected subcutaneously into nude mice. Starting from 10 or 12 days after injection, tumor volumes were measured every 2 or 3 days with a caliper and calculated using the equation, volume = width × depth × length × 0.52.

**Statistical analysis**. The data were presented as mean ± s.d. or ±s.e.m. All data are from three independent experiments. Two-sided Student's $t$-test was used to calculate $P$-values. Statistical significance is displayed as $*P < 0.05$ or $^\#P < 0.05$.

**Data availability**. The data that support the findings of this study are available from the corresponding author upon request.

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

## Acknowledgements

Our work is supported in part by Ministry of Science and Technology of China (2014CB910600 and 2017YFA0205600) and National Natural Science Foundation of China (81530076, 81525022, 31371429, 81372148, 31571472, and 81502395).

## Author contributions

H.Z. and P.G. conceived this study. X.M., L.W., D.H., H.Z., P.G., J.W., Y.S., J.G., Z.Y., X. P. and A.L. designed the experiments. X.M., L.W., D.H., Y.L., D.Y., T.L., L.S., H.W., K.H., F.Y., L.H., S.X., Z.L. and K.L. performed the experiments. F.L. and D.Z. analyzed structure information. X.M., L.W., D.H., H.Z. and P.G. wrote the paper. All the authors read and approved the manuscript.

## Additional information

**Competing interests:** The authors declare no competing financial interests.

