## [Peer Review File · Nature Communications]

Reviewers' Comments:

Reviewer #1:

Remarks to the Author:

In this study, the authors have investigated the role of polo like kinase 1 (Plk1) in biosynthesis in cancer cells. Plk1 is a serine/threonine kinase and a key regulator of mitosis and cell cycle progression. It is now known that besides cell cycle regulation Plk1 plays other important roles in regulating metabolism and cancer cell metastasis. This study suggests that Plk1 activates pentose phosphate pathway (PPP) in cancer cells by directly phosphorylating glucose-6-phosphate dehydrogenase (G6PD), a rate limiting enzyme of PPP. Activation of G6PD by Plk1 increase PPP flux to produce NADPH and ribose via glucose metabolism. The authors further demonstrate that the activation of G6PD by Plk1 is critical for cell cycle progression and cancer cell growth. The authors have produced results by utilizing both in vitro cancer cells and in vivo xenograft mouse model. The authors have made an attempt to investigate and establish the coordination between cell cycle progression and metabolic reprogramming with a common link of Plk1 in these two cellular pathways important for cancer progression.

Overall, the study is interesting and the manuscript is written and presented well. The results are presented nicely with necessary statistical analysis and discussed well in the manuscript. However, the following concerns need to be addressed by the authors.

Specific comments:

1. Some recent studies also investigated the role of Plk1 in cancer cell metabolism. These studies should be discussed in this manuscript.
2. Authors have used Plk1 inhibitor BI2536 in in vitro studies, however, they have not used this inhibitor for in vivo studies. It would be interesting to see if BI2536 has any effect on G6PD in xenograft model.
3. What was the concentration used for Plk1 inhibitor BI2536? This should be given either in fig 1 legend or on the figure itself.
4. Figure 1a. Western blot for G6PD is not convincing. In the result section, the authors suggest that there was no variation G6PD protein expression during cell cycle progression. However, G6PD expression seems to be downregulated in blot in G2/M phase compared to G1 and S phase. This downregulation could negatively affect activity of G6PD which is shown significantly increased in G2/M phase in fig. 1b. Better blot should be provided in this regard, or authors should provide densitometry data of blots normalized with loading control.
5. Fig 2a. There is a variation in beta-actin blot. It should be replaced by a better blot.
6. In some figures, the stars (*) on the graphs are not in proper place. Please correct that wherever required such as in fig. 1h, 1k, 2d and suppl 4d.
7. Suppl fig 1c, scale bar and magnification should be provided on the immunofluorescence images. In the same figure, nucleus stained with DAPI are not clearly visible. Authors may provide high-power images for the same.
8. Sppl figure 2h, the 3rd figure of phosphor-site S296 is very fuzzy and not readable.
9. Authors should more carefully proofread for grammatical mistakes. Ex. Legend to suppl fig 2(a-b). Hep3B cells "were infection" with virus, should be corrected to "were infected".

Reviewer #2:

Remarks to the Author:

Ma et al. report on a series of careful experiments designed to investigate the effects of Plk-1 on the pentose phosphate pathway during cell cycle progression. They show that in its role in activating G6PD through phosphorylation, and thereby G6PD's dimerization, Plk-1 directly affects glucose flux through the PPP and thus the rate of biosynthesis mediated by ribose-5-P production. The results provide very useful mechanistic insight into an important potential drug target that might impact a range of cancers. The results are impressive and appear to make a compelling story. To this reviewer, the experiments are well done and on the whole reasonably well described.

One question I had was why the Plk1 knockdown experiments (Fig 1n) decreased the lactate production from PPP by about 1/3. I might have expected that the effect would be somewhat larger. Maybe there is another regulator of G6PD?

I would also appreciate seeing some of the raw NMR and LC-MS data used to generate the various plots in the Figures. These data (such as a 2D NMR HSQC spectrum, chemical shift information, LC retention time, accurate mass m/z, etc.) should be added to the Supplementary Information section. For example, a number of ¹³C labeled nucleotides, as well as ¹³C labeled lactate are described, but no chemical shift assignments are given, and thus it is impossible for an interested reader to verify that proper identification was made, or to make an assessment of the data quality in terms of signal to noise. To this point, how many cells were used in the LC-MS and NMR measurements? This latter information should be provided in the Methods Section.

A related point is that the quantitation of metabolites using HSQC spectra is complicated by the fact that the metabolites can have different relaxation times. How did the authors account for this?

Overall, an impressive study worthy of publication after the above issues are addressed.

Reviewer #3:

Remarks to the Author:

In the manuscript by Ma et. al., the authors identified that PLK1 phosphorylates G6PD and promotes its dimerization, which leads to the upregulation of G6PD activity. PLK1-mediated G6PD activation is important for glucose flux through the pentose phosphate pathway to sustain nucleotide biosynthesis. Depletion of PLK1 or G6PD inhibits cell cycle progression in vitro and suppresses tumor growth in vivo. The novel of the manuscript resides in the identification of PLK1-mediated G6PD phosphorylation and its regulation on enzyme dimerization. The findings provide additional molecular insight into the coordination between cell cycle and metabolism programs in cancer cells. However, several major concerns remain to be addressed.

1. The authors showed the cell cycle-dependent changes in NADPH level and G6PD activity.

However, the authors didn't provide solid evidence that G6PD activity change is the cause for the fluctuation of NADPH level during cell cycle progression. What about the activity of other NADPH generating pathways, such as MTHFD1 and malic enzyme?

2. The authors showed the regulation of NADPH level by cell cycle and PLK1. On the other hand, NAC has minimal rescue effect upon PLK1 depletion. The authors should show whether the change in NADPH level is correlated with changes in GSH/GSSG and cellular ROS level.

3. Based on the description in Fig.1 legend, the cells were changed into C-13 labeling medium one hour before the cells were released from HU block into G1, S or G2/M phase. This means the labeling times for the samples in Fig.2K. Please clarify the experimental details. In addition, R5P is

synthesized from glucose through both oxidative and non-oxidative pentose phosphate pathway. The authors should provide the flux data for all major glucose metabolism pathways and glucose uptake data to demonstrate whether the changes in R5P is the consequence of change in G6PD activity, general decrease in glycolysis activity or suppression of glucose uptake. Along similar line, the authors should show additional flux data on glycolysis and pentose phosphate pathway from the 2C13-glucose labeling experiment and measure overall lactate production upon PLK1 knockdown or overexpression.

4. No detailed description of the measurement of mitotic time with live cell imaging was provided in Methods or Figure legends. Is the data in Fig.4a indicative of the duration of M phase? If the major function of G6PD is to provide precursor for nucleotide biosynthesis, the most affected cell cycle should be S phase. Why is G2/M phase mostly affected?

5. PLK1 can affect the progression G2/M phase through its well-known effect on cytokinesis which may explain G2/M arrest upon PLK1 knockdown in Fig.4D. However, it's intriguing why nucleoside could rescue the G2/M arrest and decrease in cell proliferation upon PLK1 knockdown. Does nucleoside also rescue the block of cytokinesis upon PLK1 depletion?

6. The authors should provide the data for cells infected with control shRNA (NTC in Fig.4A) as additional control in Fig.4C.

Point-by-point response to the comments

First of all, we appreciate greatly the constructive comments and suggestions from the reviewers. Accordingly, we have now performed additional new experiments and addressed all the concerns and comments. Our point-by-point responses are appended below.

For the reviewer's convenience, we have appended in this file all the revised figures, which we labeled as **Fig. R1** to **Fig. R11** and **Table R1** to **R3** and **Appendixes RI-RII**.

Reviewers' comments:

Reviewer #1 (Remarks to the Author):

In this study, the authors have investigated the role of polo like kinase 1 (Plk1) in biosynthesis in cancer cells. Plk1 is a serine/threonine kinase and a key regulator of mitosis and cell cycle progression. It is now known that besides cell cycle regulation Plk1 plays other important roles in regulating metabolism and cancer cell metastasis. This study suggests that Plk1 activates pentose phosphate pathway (PPP) in cancer cells by directly phosphorylating glucose-6-phosphate dehydrogenase (G6PD), a rate limiting enzyme of PPP. Activation of G6PD by Plk1 increase PPP flux to produce NADPH and ribose via glucose metabolism. The authors further demonstrate that the activation of G6PD by Plk1 is critical for cell cycle progression and cancer cell growth. The authors have produced results by utilizing both in vitro cancer cells and in vivo xenograft mouse model. The authors have made an attempt to investigate and establish the coordination between cell cycle progression and metabolic reprogramming with a common link of Plk1 in these two cellular pathways important for cancer progression.

Overall, the study is interesting and the manuscript is written and presented well. The results are presented nicely with necessary statistical analysis and discussed well in the manuscript. However, the following concerns need to be addressed by the authors.

Response: We are grateful for the reviewer's comments that well summarized the major findings and significance of our study. Indeed, we discover here an interesting coordination between cell cycle and metabolism that is mediated by Plk1. This discovery is novel and of significance. Meanwhile, we have performed additional experiments to address the reviewer's concerns and comments. We thank the reviewer for helping us to improve the manuscript substantially.

Specific comments:

Comments-1-1

1. Some recent studies also investigated the role of Plk1 in cancer cell metabolism. These studies should be discussed in this manuscript.

Response: We appreciate the reviewer's suggestion to discuss further recent studies that link Plk1 to cancer cell metabolism. In the original manuscript, we've already discussed the important literature by Li Z et al who reported that PTEN phosphorylation by Plk1 leads to its inactivation, and subsequent activation of the PI3K pathway and enhanced aerobic glycolysis (Li Z, et al., 2014). Following the reviewer's suggestion, in the Introduction section of the revised manuscript, we cited another recently published paper reporting knockdown of Plk1 alters metabolic regulation in melanoma (Gutteridge, R. E., et al., 2017). Moreover, in the Discussion section of the revised manuscript, we further discussed the report by Shao C et al that BI2536 treatment inhibits metformin-induced glycolysis and glutamine anaplerosis. (Shao C, et al., 2015). Nevertheless, it's worthwhile to note that, while those studies linked Plk1 to metabolic regulation in cancer cells, our study demonstrate for the first time that Plk1 coordinates biosynthesis during cell cycle progression by directly activating G6PD and pentose phosphate pathway.

Comments-1-2

2. Authors have used Plk1 inhibitor BI2536 in in vitro studies, however, they have not used this inhibitor for in vivo studies. It would be interesting to see if BI2536 has any effect on G6PD in xenograft model.

Response: Following this suggestion, we performed additional experiments to examine the effect of Plk1 inhibitor BI2536 on G6PD *in vivo* using xenograft model. Consistent with the data in the original manuscript, overexpression of G6PD promoted tumor growth *in vivo*, which were abolished by BI2536 treatment (**Fig. R1a**). Using the tumor tissues, we further confirmed that BI2536 treatment suppressed G6PD activity *in vivo* (**Fig. R1b**) while it exerted no effect on G6PD protein levels (**Fig. R1c**). These data further demonstrated that BI2536 suppressed G6PD activity *in vivo* to retard tumor growth. However, it's worthwhile to point out that BI2536 might also suppress tumor growth via mechanisms not related to G6PD enzyme activity since the inhibition was observed in both control and G6PD overexpression groups. We believe this to be no surprise considering the known effects of Plk1 on mitosis and so on. These new data were incorporated into the revised manuscript (**Fig. S5d-5f in the revised manuscript**).

Fig. R1. Plk1 inhibitor BI2536 suppresses G6PD activity and tumor growth *in vivo*. (a-c) Equal numbers of HeLa cells stably expressing empty vector (EV) or G6PD were subcutaneously injected into Balb/c nude mice (n=5 for each group). Vehicle (saline) or BI2536 (35 mg/kg body weight) were injected intraperitoneally one day before cell inoculation and subsequently twice a week. Tumor volumes were measured starting from 8 days after cell inoculation (a). Tumors were extracted followed by G6PD activity measurement (b) and protein analysis (c) at the end of the experiment. *P<0.05 as compared between indicated groups by two-sided student's t-test; values were represented as the mean±s.d. (see also Fig. S5d-5f in the revised manuscript).

Comments-1-3

3. What was the concentration used for Plk1 inhibitor BI2536? This should be given either in fig 1 legend or on the figure itself.

Response: The concentration used for Plk1 inhibitor BI2536 was 1 μM. We have provided this information in the Figure 1 legend and on the Figure 1f itself.

Comments-1-4

4. Figure 1a. Western blot for G6PD is not convincing. In the result section, the authors suggest that there was no variation G6PD protein expression during cell cycle progression. However, G6PD expression seems to be downregulated in blot in G2/M phase compared to G1 and S phase. This downregulation could negatively

affect activity of G6PD which is shown significantly increased in G2/M phase in fig. 1b. Better blot should be provided in this regard, or authors should provide densitometry data of blots normalized with loading control.

Response: To make this clear, we have actually performed more than three biological replicates of HeLa cells to examine the protein levels of G6PD and other PPP enzymes during cell cycle progression. As you may see from the blot data below, there was indeed no significant variation in G6PD protein expression during cell cycle progression (**Fig. R2**). To avoid confusions, we have followed the reviewer's suggestion to replace the figure by a better blot (**blot b**) in the revised manuscript (**Fig. 1a in the revised manuscript**).

Fig. R2. The protein levels of oxidative pentose phosphate pathway enzymes were not changed during cell cycle progression. (a-c) HeLa cells were synchronized to G1, S and G2/M phases. Cells were harvested and subjected to determine protein levels by western blot as indicated. Three independent biological replicates were performed. β -actin served as the loading control.

Comments-1-5

5. Fig 2a. There is a variation in beta-actin blot. It should be replaced by a better blot.

Response: We appreciate the reviewer for pointing this out. We have replaced the beta-actin blot by a better one (**blot c**) since we had performed three biological replicates (**Fig. R3 or Fig. 2a in the revised manuscript**).

Fig. R3. The western blot results of G6PD and Plk1 in different HeLa cells as indicated. (a-c) HeLa cells stably expressing NTC or shG6PD were infected with viruses expressing empty vector (EV) or Plk1. Cells were harvested and subjected to western blot. Three independent biological replicates were performed. β-actin served as the loading control.

Comments-1-6

6. In some figures, the stars () on the graphs are not in proper place. Please correct that wherever required such as in fig. 1h, 1k, 2d and suppl 4d.*

Response: We apologize for this error that leads to confusion. In the revised figures, the asterisks were positioned in the right place (Please see **Fig. 1h, 1k, 2d and Fig. S4d in the revised manuscript**).

Comments-1-7

7. Suppl fig 1c, scale bar and magnification should be provided on the immunofluorescence images. In the same figure, nucleus stained with DAPI are not clearly visible. Authors may provide high-power images for the same.

Response: We thank the reviewer for picking up this detail. Now the scale bar and magnification were provided and the DAPI images were refined (**Fig. R4 or Fig. S1c in the revised manuscript**).

Fig. R4. Immunofluorescence staining of pH3Ser10 in synchronized HeLa cells treated with BI2536 or vehicle. HeLa cells were synchronized into G1 phase and released to fresh DMEM for 0, 5, 10, 12 hours. BI2536 or DMSO was added 1 hour before releasing. Cells were subjected to immunofluorescence assay to stain mitotic marker pH3Ser10 (see also Fig. S1c in the revised manuscript).

Comments-1-8

8. Suppl figure 2h, the 3rd figure of phosphor-site S296 is very fuzzy and not readable.

Response: Following the reviewer's suggestion, the image of phosphor-site S296 was replaced by a better one with higher resolution (Fig. R5 or Fig. S2h in the revised manuscript).

phospho-site S296

Spectrum from G6FD-2.wiff (sample 1) - Sample003, Experiment 22, *TOF MS*2 (100 - 1600) from 31.319 min
Precursor: 975.1 Da

Fig. R5. LC-MS/MS identified phosphorylation sites of Plk1 on G6PD. The phosphorylation site (S296) of Plk1 on G6PD was identified by mass spectrometry (see also **Fig. S2h** in the revised manuscript).

Comments-1-9

9. Authors should more carefully proofread for grammatical mistakes. Ex. Legend to suppl fig 2(a-b). Hep3B cells “were infection” with virus, should be corrected to “were infected”.

Response: We have carefully proofread the entire manuscript and corrected as much as we could the typo errors and grammatical mistakes in the revised manuscript. Thank you for all your constructive comments and suggestions that helped us to improve the manuscript significantly.

Reviewer #2 (Remarks to the Author):

Ma et al. report on a series of careful experiments designed to investigate the effects of Plk-1 on the pentose phosphate pathway during cell cycle progression. They show that in its role in activating G6PD through phosphorylation, and thereby G6PD's dimerization, Plk-1 directly affects glucose flux through the PPP and thus the rate of biosynthesis mediated by ribose-5-P production. The results provide very useful mechanistic insight into an important potential drug target that might impact a range of cancers. The results are impressive and appear to make a compelling story. To this reviewer, the experiments are well done and on the whole reasonably well described.

Response: We appreciate the reviewer for the positive comments pointing out the significance and novelty of our study.

Comments-2-1

One question I had was why the Plk1 knockdown experiments (Fig 1n) decreased the lactate production from PPP by about 1/3. I might have expected that the effect would be somewhat larger. Maybe there is another regulator of G6PD?

Response: We thank the reviewer for raising this relevant question. We understand that the reviewer might expect a larger decrease in the level of lactate production following Plk1 knockdown. However, considering that lactate is the final product of glycolysis, the decrease of about 1/3 in its production by Plk1 knockdown is really significant. Moreover, the shRNA-mediated knockdown can't deplete the gene expression of Plk1 completely, thus potentially limiting its effects on G6PD and

downstream functions. Indeed, as the reviewer has pointed out, there exist other molecules such as P53 in regulation of G6PD and PPP pathway (Already discussed in the manuscript). Nevertheless, based on our multiple lines of evidence, Plk1 markedly regulates G6PD activity by interacting with and phosphorylating G6PD.

Comments-2-2

I would also appreciate seeing some of the raw NMR and LC-MS data used to generate the various plots in the Figures. These data (such as a 2D NMR HSQC spectrum, chemical shift information, LC retention time, accurate mass m/z, etc.) should be added to the Supplementary Information section. For example, a number of ¹³C labeled nucleotides, as well as ¹³C labeled lactate are described, but no chemical shift assignments are given, and thus it is impossible for an interested reader to verify that proper identification was made, or to make an assessment of the data quality in terms of signal to noise. To this point, how many cells were used in the LC-MS and NMR measurements? This latter information should be provided in the Methods Section.

Response: Per the request of the reviewer, we have provided the NMR and LC-MS raw data as well as the qualitative and quantitative methods (**Fig. R6, Table R1-R2 and Appendix RI-RII**). **Table R1 and R2** were also included in the Supplementary Information in the revised manuscript (**Table s1 to s2 in the Supplementary Information**).

For NMR data: We provided the metabolite assignments of the 2D HSQC spectra in **Fig. R6a and 6b**. Detailed chemical shift information of individual metabolite can be found in **Table R1**. For space consideration, all assignment-related NMR spectra and HSQC spectra of standards and cell extracts and the raw data of the peak volumes (normalized by cell wet weight) of the cross-peaks in 2D HSQC spectra are provided in the **Appendix RI**.

For LC-MS data: We provided retention time and accurate m/z information of each metabolite in **Table R2**. For space consideration, the raw LC-MS data as well as qualitative and quantitative methods were provided in the **Appendix RII**.

The cells we used: For NMR measurements, the peak volumes of the cross peaks were normalized to cell wet weights. The cell numbers are estimated to be roughly 3×10^7 per 100 mg wet weight. For most samples, 80-100 mg of wet weight cells (roughly $2.5-3 \times 10^7$ cells/sample) were used. For LC-MS, the cell number we used is 5×10^6 . We have provided this cell number information in the Method Section in the revised manuscript.

Fig. R6. The HSQC spectra and assignments (a) HSQC spectrum of 2- ^{13}C glucose labeled sample (b) HSQC spectrum of U- ^{13}C glucose labeled sample.

Table R1: Chemical shift of each metabolite

Compound	Group	Structure	$\delta(^1\text{H})/\text{ppm}$	$\delta(^{13}\text{C})/\text{ppm}$
R5P	C1		5.24	104.08
CXP	C1		6.15	99.47
UXP	C1		5.98	105.49
GXP	C1'		5.95	89.53
AXP	C1'		6.14	89.79
Lactate	C3		1.32	23.08
	C2		4.10	72.08
G6P	C2		3.27	76.94
Pyruvate	C3		2.35	29.80

See also Table s1 in the revised manuscript.

Table R2: Retention time and accurate m/z of each metabolite

Metabolite	RT	M/Z
R5P	5.701	299.0119 +/- 0.0025Da
AMP	5.046	346.056 +/- 0.010Da
dTMP	4.040	321.049 +/- 0.010Da
dCMP	5.517	306.050 +/- 0.005Da

See also Table s2 in the revised manuscript.

Comments-2-3

A related point is that the quantitation of metabolites using HSQC spectra is complicated by the fact that the metabolites can have different relaxation times. How did the authors account for this?

Overall, an impressive study worthy of publication after the above issues are addressed.

Response: As the reviewer pointed out, 2D cross-peak intensities (or volumes) are influenced by a greater number of variables (e.g., uneven excitation, non-uniform relaxation, evolution times, mixing times, etc.), which makes it difficult to translate peak intensities into absolute metabolite concentrations. Lewis et al proposed a FMQ approach (FMQ: fast metabolite quantification). For each pre-identified target metabolite resonance, the non-uniform behavior including relaxation effects for cross-peaks of individual metabolites are expected to be the same, considering the fact that all cell extracts were under the same treatment and the same NMR acquisition parameter set were applied (Lewis, et al., 2007). According to this FMQ method, a standard curve can be constructed for each metabolite by regressing absolute peak intensities from the concentration reference samples with their known concentrations, and accurate (technical error 2.7%) molar concentrations can be determined. In this study, we followed the general guideline of FMQ method, except that we are not trying to determine the absolute metabolite concentration. Instead, we are interested in the relative concentration changes of the same metabolite in various cell samples. Therefore, the good linear correlation between resonance intensity and corresponding concentration that was demonstrated in Lewis paper, set a solid foundation for characterizing the metabolite concentration changes by their relative intensity changes in 2D NMR signals. Here we utilize the changes in 2D cross-peak intensities to quantify their concentration changes. For metabolites like ATP, ADP and AMP, some of the resonances are identical, which may bring in quantitation complexity if they have different relaxation times. In this study, we carefully measured the T1 relaxation times of AXPs, CXPs, UXPs and GXPs, and assessed the potential errors associated with relaxation delay. For detailed information please see Appendix RII and Table R3 (Table s3 in the revised Supplementary Information).

Table R3: T1 experiments and calculation

298k, 10mM	ATP	ADP	AMP	CTP	CDP	CMP	GTP	GDP	GMP	UTP	UDP	UMP
T1	8.70	7.85	8.79	7.97	6.88	6.86	5.69	7.30	7.65	7.29	7.93	8.57
M(t)/M(0), * t=1.5s	0.16	0.17	0.16	0.17	0.20	0.20	0.23	0.19	0.18	0.19	0.17	0.16
SD		0.01			0.01			0.03			0.01	

* M(t) and M(0) are the magnetization at the time t and equilibrium state, t: relaxation delay. See also Table s3 in the revised manuscript.

Reviewer #3 (Remarks to the Author):

In the manuscript by Ma et. al., the authors identified that PLK1 phosphorylates G6PD and promotes its dimerization, which leads to the upregulation of G6PD activity. PLK1-mediated G6PD activation is important for glucose flux through the pentose phosphate pathway to sustain nucleotide biosynthesis. Depletion of PLK1 or G6PD inhibits cell cycle progression in vitro and suppresses tumor growth in vivo. The novel of the manuscript resides in the identification of PLK1-mediated G6PD phosphorylation and its regulation on enzyme dimerization. The findings provide additional molecular insight into the coordination between cell cycle and metabolism programs in cancer cells. However, several major concerns remain to be addressed.

Response: We appreciate the reviewer for pointing out the novelty and significance of this manuscript. Indeed, our study provides additional molecular insight into the coordination between cell cycle and metabolism in cancer cells. Meanwhile, we are also grateful for his/her multiple critical comments which have helped us to substantially improve the manuscript. We have addressed all the comments/concerns below and revised the manuscript.

Comments-3-1

1. The authors showed the cell cycle-dependent changes in NADPH level and G6PD activity. However, the authors didn't provide solid evidence that G6PD activity change is the cause for the fluctuation of NADPH level during cell cycle progression. What about the activity of other NADPH generating pathways, such as MTHFD1 and malic enzyme?

Response: Indeed, there exist multiple NADPH generating pathways, which involve many other metabolic enzymes such as MTHFD1, ME (malic enzyme) and IDH (Isocitrate dehydrogenase). To address the reviewer's questions experimentally, we first measured the protein levels of these NADPH generating enzymes in synchronized cells or in cells with Plk1 overexpression/knockdown, and detected no significant variations in protein levels of MTHFD1, ME1 or IDH2 (**Fig. R7a-c**). Next, we measured the activity of ME and IDH, as a result, no significant activity changes were observed for these enzymes in synchronized cells or in cells with different Plk1 expression (**Fig. R7d-i**). These results, together with our observation that G6PD activity fluctuates during cell cycle progression or with Plk1 protein expression, further suggest that G6PD is likely the enzyme whose activity changes represent the cause for the fluctuation of NADPH level during cell cycle progression. Thus, we further synchronized HeLa cells expressing NTC or shG6PD and measured NADPH

levels. The results showed that the increment of NADPH during cell cycle progression was significantly blocked by shG6PD (Fig. R7j), further confirming that G6PD is important for the fluctuation of NADPH during cell cycle progression. Of note, because no commercial kit is currently available to us, unfortunately, we were unable to measure the activity of MTHFD1.

Fig. R7. The protein levels and activities of other NADPH generating enzymes during cell cycle progression. (a-c) Protein levels of the major NADPH generating enzymes were determined by western blot in HeLa cells synchronized into G1, S and G2/M phases (a), or in HeLa cells overexpressing Plk1 (b) or expressing tet-inducible shPlk1 (c). (d-i) Enzyme activities of malic enzyme (ME) and Isocitrate dehydrogenase (IDH) were measured in HeLa cells synchronized into G1, S and G2/M phases (d-e), or in HeLa cells overexpressing Plk1 (f-g), or expressing tet-inducible shPlk1 (h-i). (j) HeLa cells expressing NTC or shG6PD were synchronized into G1, S and G2/M phases. Cells were harvested and NADPH levels were measured by commercial kit. * $P < 0.05$ as compared to G1-NTC group, # $P < 0.05$ as compared to corresponding NTC group, by two-sided student's t-test. ns: not significant. Values represent the mean \pm s.d. β -actin serves as the loading control.

Comments-3-2

2. The authors showed the regulation of NADPH level by cell cycle and PLK1. On the other hand, NAC has minimal rescue effect upon PLK1 depletion. The authors should show whether the change in NADPH level is correlated with changes in GSH/GSSG and cellular ROS level.

Response: As suggested by the reviewer, we performed additional assays to measure GSH and ROS levels in HeLa cells with Plk1 overexpression or knockdown. The results showed that, consistent with the changes of NADPH, Plk1 expression affected the levels of GSH and ROS as well (**Fig. R8 or Fig. S1g-1k in the revised manuscript**), indicating that, by regulating G6PD-mediated biomass synthesis, Plk1 had an effect on cellular redox status via NADPH/GSH/ROS. However, the role of Plk1 on PPP pathway and biosynthesis is obviously not limited to NADPH. Thus, while we did observe that NAC exerted some rescue effects upon Plk1 depletion, the

effects of nucleotides were more obvious in our experimental systems. NAC alone was not very efficient. We actually did all subsequent rescue experiments with Nuc and NAC mixture (Fig. 4 and Fig. S4 in the original manuscript).

Fig. R8. Plk1/G6PD regulates cellular GSH and ROS levels (a-c) GSH levels were measured in HeLa cells overexpressing Plk1 (a), or expressing shPlk1 (b) or in Plk1 overexpressing HeLa cells with further G6PD knockdown (c). (d-e) ROS levels were measured in HeLa cells expressing tet-inducible shPlk1 (d) or in HeLa cells overexpressing Plk1 (e). * $P < 0.05$ as compared to EV or NTC group by two-sided student's t-test; values represent the mean \pm s.d. See also Fig. S1g-1k in the revised manuscript)

Comments-3-3

3. Based on the description in Fig.1 legend, the cells were changed into C-13 labeling medium one hour before the cells were released from HU block into G1, S or G2/M phase. This means the labeling times for the samples in Fig.2K. Please clarify the experimental details. In addition, R5P is synthesized from glucose through both oxidative and non-oxidative pentose phosphate pathway. The authors should provide the flux data for all major glucose metabolism pathways and glucose uptake data to demonstrate whether the changes in R5P is the consequence of change in G6PD

activity, general decrease in glycolysis activity or suppression of glucose uptake. Along similar line, the authors should show on glycolysis and pentose phosphate pathway from the 2C13-glucose labeling experiment and measure overall lactate production upon PLK1 knockdown or overexpression.

Response: The reviewer is right about the labeling time of the samples in Fig. 1k. Following the reviewer's suggestion, we have provided more experimental details in the legend of Fig. 1k in the revised manuscript. Briefly, HeLa cells were synchronized by double HU block which was described in Methods section of the manuscript. As shown in **Fig. R9a**, when the cells were treated for the second round of HU block for 11 hours, i.e. 1 hour before releasing, U-¹³C glucose was added into the cell culture medium and further culture for 1 hour and then harvested for G1 phase cell analysis. For cell samples of S and G2/M phases, after the regular second round of HU block for 12 hours, cells were released into fresh HU-free medium and further cultured for 4 hours and 9 hours, respectively, followed by supplementation of U-¹³C glucose and 1 hour culture before harvesting for S and G2/M phases cell analysis.

As the reviewer has pointed out, R5P is synthesized from glucose through both oxidative and non-oxidative pentose phosphate pathway. Our data indicate that Plk1 expression may affect the production of R5P from ¹³C-labeled glucose, however, these data could not distinguish whether the production of R5P is from the oxidative or non-oxidative pentose phosphate pathway. This is very complicated. As shown in **Fig. R9b**, there are multiple combinations of R5P generation when tracing the carbon using 2-¹³C glucose via general glycolysis or pentose phosphate pathways. While the oxidative PPP is simple and more straightforward, the non-oxidative one could generate many potential products, whose signal via ¹³C-labeling tracing could be very weak. Moreover, both the oxidative and non-oxidative PPP could potentially generate the same 1-¹³C-R5P product from 2-¹³C-glucose. As a result, it is very difficult for us to distinguish R5P from the two pathways even using 2-¹³C-glucose labeling experiments.

However, following the reviewer's comments, we performed additional experiments to further clarify this important point. We knocked down G6PD in HeLa cells overexpressing Plk1 to suppress the oxidative PPP activity but not the non-oxidative one, and performed U-¹³C glucose metabolic flux assay. As a result, Plk1 overexpression significantly enhanced R5P production, which was abolished by shG6PD (**Fig. R9c**). Consistently, shG6PD also significantly abolished the Plk1 overexpression-induced increase in the production of GXP, UXP, CXP and AXP (**Fig. R9c**). Collectively, these data demonstrate that G6PD-mediated oxidative pentose phosphate pathway was the major cause for changes in R5P and subsequent

nucleotides levels regulated by Plk1. We have included these new data in the revised manuscript (**Fig. 1q in the revised manuscript**).

C
Fig. R9. Plk1 promotes oxidative pentose phosphate pathway. (a) The time table for preparing G1, S and G2/M synchronized cell samples in U-¹³C glucose labeling experiment. (b) The ¹³C carbon flow of 2-¹³C glucose through glycolysis and PPP. (c) U-¹³C glucose metabolic flux assays were performed in Plk1 overexpressing HeLa cells with G6PD knockdown by NMR. *P<0.05 as compared to corresponding EV-NTC group by two-sided student's t-test; values were represented as the mean± s.e.m. (See also **Fig. 1q** in the revised manuscript).

Following the reviewer's suggestion, we also measured changes in glucose uptake as well as the levels of some glycolytic metabolites in HeLa cells and found that Plk1 overexpression enhanced glucose uptake and the production of lactate, G6P and pyruvate, indicating that Plk1 did have effects on glycolysis in general (**Figure R10a-10f**). This is consistent with a previous report by Li Z et al, showing that Plk1 regulates glycolysis in cancer cells via targeting PTEN/PI3K pathway (Li et al., 2014). However, our study showed clearly that Plk1 activates G6PD and oxidative pentose phosphate pathway to promote biosynthesis and cancer progression. We have included these new data in the revised manuscript (**Fig. S10-1t** in the revised manuscript).

Fig. R10. Plk1 regulates glucose uptake and glycolysis. (a-d) Glucose uptake or overall lactate production was measured in HeLa cells overexpressing Plk1 (a, c) or expressing tet-inducible shPlk1 (b, d). (e-f) ^{13}C -incorporated G6P and pyruvate were measured in $2\text{-}^{13}\text{C}$ glucose metabolic flux assay by NMR in HeLa cells overexpressing Plk1 (e) or expressing tet-inducible shPlk1 (f). * $P < 0.05$ as compared to corresponding EV or $-\text{tet}$ group by two-sided student's t-test; values were represented as the mean \pm s.d. or s.e.m. (See also Fig. S10-1t in the revised manuscript).

Comments-3-4

4. No detailed description of the measurement of mitotic time with live cell imaging

was provided in Methods or Figure legends. Is the data in Fig.4a indicative of the duration of M phase? If the major function of G6PD is to provide precursor for nucleotide biosynthesis, the most affected cell cycle should be S phase. Why is G2/M phase mostly affected?

Response: We thank the reviewer for pointing this out. We have now provided detailed description of the measurement of mitotic time with live cell imaging in Method section and figure legends in revised manuscript. Yes, the data in **Fig. 4a** are indicative of the duration of M phase.

As a matter of fact, our results showed that both S phase and G2/M phases were affected by Plk1 and that addition of Nuc and NAC could significantly rescue both the S phase and G2/M phase cell population suppressed by shPlk1 (**Fig. 4d**). Indeed, we confirmed repeatedly that G6PD activity starts to increase during S phase, but reaches the peak level at G2/M instead of S phase, we don't know why exactly. It's likely that, during cell cycle progression, the biosynthesis process continues beyond S phase, as Atilla-Gokcumen documented recently that the lipid composition of cells in S phase differs drastically from cells in cytokinesis (Atilla-Gokcumen, et al., 2014). We believe the reviewer raised here a very stimulating question, therein potentially lies paradigm-shifting mechanisms and answers for the control of cell cycle progression.

Comments-3-5

5. PLK1 can affect the progression G2/M phase through its well-known effect on cytokinesis which may explain G2/M arrest upon PLK1 knockdown in Fig.4D. However, it's intriguing why nucleoside could rescue the G2/M arrest and decrease in cell proliferation upon PLK1 knockdown. Does nucleoside also rescue the block of cytokinesis upon PLK1 depletion?

Response: Again, this is a very stimulating question. At this moment, we don't have any evidence demonstrating that nucleosides also rescue the block of cytokinesis upon Plk1 depletion. However, it's reasonable for us to believe that, when the knockdown effect of Plk1 is not complete with shPlk1 and so is its blocking effect on cytokinesis, the addition of nucleosides could partially rescue the S and G2/M arrest and cell proliferation, as we have observed in **Fig. 4d-4e**.

Comments-3-6

6. The authors should provide the data for cells infected with control shRNA (NTC in Fig.4A) as additional control in Fig.4C.

Response: We thank the reviewer for the suggestion. Accordingly, we performed new experiments to include the NTC-EV group as an additional control. The results showed that the cell number of NTC-EV group was similar to that of the wild-type group (**Fig. R10**). We also replaced the **Fig. 4c** in the revised manuscript.

Fig. R11. The effects of G6PD mutation on cell proliferation. HeLa cells stably expressing NTC or shG6PD were further infected with viruses expressing empty vector (EV) or Flag-G6PD wild-type or its mutants as indicated. Cell were treated with vehicle or Nuc mix and NAC supplementation. Cell numbers were counted 4 days after treatment. * $P < 0.05$ as compared between indicated groups by two-sided student's t-test; values were represented as the mean \pm s.d. (See also **Fig. 4c** in the revised manuscript).

References:

1. Li Z, *et al.* Plk1 Phosphorylation of PTEN Causes a Tumor-Promoting Metabolic State. *Molecular and cellular biology* 34, 3642-3661 (2014).
2. Shao C, Ahmad N, Hodges K, Kuang S, Ratliff T, Liu X. Inhibition of polo-like kinase 1 (Plk1) enhances the antineoplastic activity of metformin in prostate cancer. *Journal of Biological Chemistry* 290, 2024-2033 (2015).
3. Gutteridge REA, Singh CK, Ndiaye MA, Ahmad N. Targeted knockdown of polo-like kinase 1 alters metabolic regulation in melanoma. *Cancer letters* 394, 13-21 (2017).
4. Lewis, I. A., *et al.* Method for determining molar concentrations of metabolites in complex solutions from two-dimensional H-1-C-13 NMR spectra. *Analytical Chemistry* 79(24): 9385-9390 (2007).
5. Atilla-Gokcumen GE, *et al.* Dividing cells regulate their lipid composition and localization. *Cell* 156, 428-439 (2014).

Appendix RI:

NMR HSQC spectra and quantification of each metabolite.

The HSQC spectra of each metabolite standard.

R5P:

AXP:

GXP:

CXP:

UXP:

G6P:

Lactate:

Pyruvate:

T1 measurement using inversion recovery:

* $M(t)$ and $M(0)$ is the magnetization at the time t and equilibrium state, RD: relaxation delay.

The HSQC spectrum of 2-¹³C labeled sample:

The HSQC spectrum of U-¹³C labeled sample:

The normalized integrals of our raw NMR data:

For 2-¹³C glucose labeled samples (Fig. 1m-1n, Fig. S1s-1t):

the integrals of each metabolite				integrals			
No.	sample	labeling	replicates	G6P	Pyruvate	Lactate 2-C	Lactate 3-C
1	EV	2-13C glucose	1	1675580.1	6109502.76	38381215.47	44082872.93
2	EV	2-13C glucose	2	2168379.8	3239198.61	43921602.79	51533101.05
3	EV	2-13C glucose	3	2584292.6	11970263.8	95436450.84	149280575.5
4	Plk1	2-13C glucose	1	3333333.3	6736178.86	57593495.93	59086720.87
5	Plk1	2-13C glucose	2	2861279.8	5103036.88	47207158.35	48849240.78
6	Plk1	2-13C glucose	3	2703689.6	16821883	60035623.41	99950381.68
7	tet-	2-13C glucose	1	6763681.6	19063018.2	74338498.21	141358760.4
8	tet-	2-13C glucose	2	10019854	12159380.7	7301854.305	11351258.28
9	tet-	2-13C glucose	3	6121924.5	7712302.07	25910569.11	38889227.64
10	tet+	2-13C glucose	1	4073897.5	14576877.2	136867330	167114427.9
11	tet+	2-13C glucose	2	7301854.3	11351258.3	31438979.96	81295081.97
12	tet+	2-13C glucose	3	5595528.5	6629369.92	45308160.78	60544457.98

For U-¹³C glucose labeled samples (Fig. 1o-1q):

No.	sample	labeling	replicates	integrals of each metabolite							
				R5P	CXP	UXP	GXP	AXP	G6P- β 1	G6P- β 2	Pyruvate
1	EV	U-13C glucose	1	8223307.5	5644197.29	143626692.5	20533849.13	45862669.2	38668936.2	17317446.81	96612765.96
2	EV	U-13C glucose	2	6185617	5953446.81	163089361.7	20935319.15	41066383	44183752.4	23572533.85	75792069.63
3	EV	U-13C glucose	3	4913659.5	11304955.5	92401524.78	26693773.82	31661372.3	19209021.6	4359656.925	101581956.8
4	Plk1	U-13C glucose	1	11207692	10237252.7	205560439.6	40848351.65	62017582.4	43098540.1	18231751.82	97198905.11
5	Plk1	U-13C glucose	2	13539234	9002098.54	181861313.9	32114963.5	48680656.9	103674725	59383516.48	106437362.6
6	Plk1	U-13C glucose	3	9159187.1	14637595.3	108704487.7	42217612.19	45245554.6	30329381.9	13461473.33	123700254
7	tet-	U-13C glucose	1	15612590	8294268.17	48984646.88	14694984.65	4491044.01	1234237462	845598771.8	1590122825
8	tet-	U-13C glucose	2	6213359.3	12064461.7	227872892.3	40394293.13	6659662.78	145633397	90307101.73	116619961.6
9	tet-	U-13C glucose	3	7137842.3	14117842.3	243784232.4	33723651.45	70627385.9	46794190.9	33002489.63	46794190.87
10	tet+	U-13C glucose	1	7661503.2	3241250.72	31148594.38	8989099.254	2294205.39	393247275	266815834.8	194394721.7
11	tet+	U-13C glucose	2	3550479.8	4764491.36	160946257.2	19781190.02	3690403.07	86603112.8	57966277.56	55186770.43
12	tet+	U-13C glucose	3	3242939.5	7649951.97	190518732	26819404.42	33437079.7	130297791	91635926.99	100355427.5
13	ntc+ev	U-13C glucose	1	19261084	26502463.1	175233990.1	45570197.04	75123152.7	68772167.5	73658867	98656403.94
14	ntc+ev	U-13C glucose	2	92721228	23960358.1	180895140.7	40195652.17	101088235	159386189	153081841.4	259872122.8
15	ntc+ev	U-13C glucose	3	46240621	19987063.4	185226390.7	39827943.08	59439844.8	94873221.2	86853816.3	131578266.5
16	ntc+ev	U-13C glucose	4	60411465	5969808.92	142305732.5	17347770.7	46182165.6	80892993.6	69231847.13	133222929.9
17	ntc+Plk1	U-13C glucose	1	21091292	37331460.7	211207865.2	56026685.39	180042135	87054775.3	85759831.46	135529494.4
18	ntc+Plk1	U-13C glucose	2	92532951	11394555.9	155673352.4	32522922.64	56932664.8	88866762.2	93767908.31	151762177.7
19	ntc+Plk1	U-13C glucose	3	122267705	143286119	13947167.14	33443342.78	71558073.7	62143059.5	65349858.36	202733711
20	ntc+Plk1	U-13C glucose	4	81168375	10680380.7	156881405.6	31054172.77	57619326.5	99636896	99232796.49	139338213.8
21	shG6PD+EV	U-13C glucose	1	12853012	25572289.2	54307228.92	25920481.93	58492771.1	26569879.5	69086746.99	77034939.76
22	shG6PD+EV	U-13C glucose	2	40897686	14158343.5	45129110.84	17028014.62	37940316.7	33892813.6	84560292.33	66624847.75
23	shG6PD+EV	U-13C glucose	3	31327607	9016073.62	42570552.15	13679754.6	50640490.8	30694478.5	79769325.15	64165644.17
24	shG6PD+EV	U-13C glucose	4	103415274	17373508.4	33091885.44	12492840.1	48998806.7	28368735.1	80659904.53	65813842.48
25	shG6PD+Plk1	U-13C glucose	1	27300000	44408974.4	184038461.5	219217948.7	1129666667	72993589.7	63701282.05	82105128.21
26	shG6PD+Plk1	U-13C glucose	2	74891753	14426546.4	120612113.4	29201030.93	49344072.2	121016753	105319587.6	82051546.39
27	shG6PD+Plk1	U-13C glucose	3	43288265	16127551	113852040.8	31028061.22	53627551	109724490	114506377.6	82875000
28	shG6PD+Plk1	U-13C glucose	4	35593264	13257772	110152849.7	27707253.89	87084196.9	210194301	202396373.1	148354922.3

Appendix RII:

LC-MS raw data and processing:

For metabolites identification via LC-MS, we first run standards for R5P, AMP, dTMP and dCMP to determine the retention time and m/z. The spectra are as below:

R5P-STD

AMP-STD

dTMP-STD

dCMP-STD

Identification information was summarized in the table below.

Metabolite	RT	M/Z
R5P	5.701	299.0119 +/- 0.0025Da
AMP	5.046	346.056 +/- 0.010Da
dTMP	4.040	321.049 +/- 0.010Da
dCMP	5.517	306.050 +/- 0.005Da

For ¹³C-labeling experiments, we take R5P for example to explain how the data was acquired and processed. For R5P, the isotope peak measured was m0, m1 and m5 (the missing isotope peak was counted as 0) (Fig. 1k in the manuscript). The spectra are as follows:

R5P (m0)

R5P (m1)

R5P (m5)

We performed relative quantification using peak area and used ISOCOR software for correcting naturally-occurring isotopes to get the isotopologue distribution. Raw data and processed data are as below:

Peak area

(1h)	G1			S			G2M		
R5P	1	2	3	1	2	3	1	2	3
M+0	11969.83	10965.66	11274.89	21861.72	20281.73	22341.63	18209.7	16586.66	16024.2
M+1	18317.9	16260.48	17936.14	26931.65	29045.73	28185.63	31154.6	30104.82	25310.13
M+2	0	0	0	0	0	0	0	0	0
M+3	0	0	0	0	0	0	0	0	0
M+4	0	0	0	0	0	0	0	0	0
M+5	1890.68	1811	1849.59	4358.67	4160.97	4672.49	5137.87	5137.49	5307.36

Isotopologue distribution

(1h)	G1			S			G2M		
R5P	1	2	3	1	2	3	1	2	3
M+0	0.37742	0.38346	0.36791	0.41937	0.38521	0.41241	0.33764	0.32281	0.34779
M+1	0.56248	0.55273	0.57121	0.49656	0.53516	0.50083	0.56607	0.576	0.53592
M+2	0	0	0	0	0	0	0	0	0
M+3	0	0	0	0	0	0	0	0	0
M+4	0	0	0	0	0	0	0	0	0
M+5	0.0601	0.06382	0.06088	0.08407	0.07963	0.08676	0.0963	0.10119	0.1163

We converted the M+5 value to percentage and the averages of the percentage values of three biological replicates were used in the bar-graph.

R5P				%	%	%		AVE	STDEV
	0.0601	0.06382	0.06088	6.01	6.382	6.088	G1	6.16	0.196173
	0.08407	0.07963	0.08676	8.407	7.963	8.676	S	8.34867	0.360062
	0.0963	0.10119	0.1163	9.63	10.119	11.63	G2/M	10.4597	1.042612

LC-MS spectra and quantification of AMP, dTMP, dCMP. (Fig. S1n in the revised manuscript).

AMP (M0)

AMP (m2)

AMP (m3)

AMP (m4)

AMP (m5)

dTMP (m0)

dTMP (m1)

dTMP (m4)

dTMP (m5)

dCMP (m0)

dCMP (m1)

dCMP (m2)

dCMP (m3)

dCMP (m4)

dCMP (m5)

We also find that, in the original Fig. S1i, we've made a mistake---used the percentage values of one column of individual samples instead of the average value of the three replicates in generating the bar-graph (yellow shade). Now it is corrected in the revised version (green shade) and the trends of these metabolite changes remain the same (Fig. S1n in the revise manuscript).

Data used for bar graph were shown in the excel table below:

replicate	1	2	3		1	2	3		AVE	STDEV
AMP 2h					%	%	%			
	0.15228	0.1533	0.15352		15.228	15.33	15.352	G1	15.3033	0.066161
	0.19213	0.17479	0.18169		19.213	17.479	18.169	S	18.287	0.873002
	0.21509	0.20448	0.23317		21.509	20.448	23.317	G2/M	21.758	1.450617
dTMP 2h					%	%	%		AVE	STDEV
	0.0665	0.070965	0.07543		6.65	7.0965	7.543	G1	7.0965	0.4465
	0.12099	0.115185	0.10938		12.099	11.5185	10.938	S	11.5185	0.5805
	0.1092	0.124435	0.13967		10.92	12.4435	13.967	G2/M	12.4435	1.5235
dCMP 2h					%	%	%		AVE	STDEV
	0.09444	0.095	0.08294		9.444	9.5	8.294	G1	9.07933	0.680695
	0.09345	0.10254	0.11444		9.345	10.254	11.444	S	10.3477	1.05263
	0.10576	0.10998	0.11862		10.576	10.998	11.862	G2/M	11.1453	0.655537
		Former						Corrected		
		AMP	dTMP	dCMP				AMP	dTMP	dCMP
	G1	15.352	7.543	8.294		G1		15.30333	7.0965	9.079333
	S	18.169	10.938	11.444		S		18.287	11.5185	10.34767
	G2/M	23.317	13.967	11.862		G2/M		21.758	12.4435	11.14533
	STDEV	0.066161	0.4465	0.680695		STDEV		0.066161	0.4465	0.680695
		0.873002	0.5805	1.05263				0.873002	0.5805	1.05263
		1.450617	1.5235	0.655537				1.450617	1.5235	0.655537

The bar graph generated:

Reviewers' Comments:

Reviewer #1:

Remarks to the Author:

The authors have adequately addressed reviewers' concerns and have revised their manuscript, accordingly. I do not have any additional concerns.

Reviewer #2:

Remarks to the Author:

The authors have answered all of my previous questions and comments.

Reviewer #3:

Remarks to the Author:

The authors have nicely addressed all my concerns and I have no further comment.

Reviewers' comments:

Reviewer #1 (Remarks to the Author):

In this study, the authors have investigated the role of polo like kinase 1 (Plk1) in biosynthesis in cancer cells. Plk1 is a serine/threonine kinase and a key regulator of mitosis and cell cycle progression. It is now known that besides cell cycle regulation Plk1 plays other important roles in regulating metabolism and cancer cell metastasis. This study suggests that Plk1 activates pentose phosphate pathway (PPP) in cancer cells by directly phosphorylating glucose-6-phosphate dehydrogenase (G6PD), a rate limiting enzyme of PPP. Activation of G6PD by Plk1 increase PPP flux to produce NADPH and ribose via glucose metabolism. The authors further demonstrate that the activation of G6PD by Plk1 is critical for cell cycle progression and cancer cell growth. The authors have produced results by utilizing both in vitro cancer cells and in vivo xenograft mouse model. The authors have made an attempt to investigate and establish the coordination between cell cycle progression and metabolic reprogramming with a common link of Plk1 in these two cellular pathways important for cancer progression.

Overall, the study is interesting and the manuscript is written and presented well. The results are presented nicely with necessary statistical analysis and discussed well in the manuscript. However, the following concerns need to be addressed by the authors.

Specific comments:

1. Some recent studies also investigated the role of Plk1 in cancer cell metabolism. These studies should be discussed in this manuscript.
2. Authors have used Plk1 inhibitor BI2536 in in vitro studies, however, they have not used this inhibitor for in vivo studies. It would be interesting to see if BI2536 has any effect on G6PD in xenograft model.
3. What was the concentration used for Plk1 inhibitor BI2536? This should be given either in fig 1 legend or on the figure itself.
4. Figure 1a. Western blot for G6PD is not convincing. In the result section, the authors suggest that there was no variation G6PD protein expression during cell cycle progression. However, G6PD expression seems to be downregulated in blot in G2/M phase compared to G1 and S phase. This downregulation could negatively affect activity of G6PD which is shown significantly increased in G2/M phase in fig. 1b. Better blot should be provided in this regard, or authors should provide densitometry data of blots normalized with loading control.
5. Fig 2a. There is a variation in beta-actin blot. It should be replaced by a better blot.
6. In some figures, the stars (*) on the graphs are not in proper place. Please correct that wherever required such as in fig. 1h, 1k, 2d and suppl 4d.

7. Suppl fig 1c, scale bar and magnification should be provided on the immunofluorescence images. In the same figure, nucleus stained with DAPI are not clearly visible. Authors may provide high-power images for the same.

8. Sppl figure 2h, the 3rd figure of phosphor-site S296 is very fuzzy and not readable.

9. Authors should more carefully proofread for grammatical mistakes. Ex. Legend to suppl fig 2(a-b). Hep3B cells "were infection" with virus, should be corrected to "were infected".

Reviewer #2 (Remarks to the Author):

Ma et al. report on a series of careful experiments designed to investigate the effects of Plk-1 on the pentose phosphate pathway during cell cycle progression. They show that in its role in activating G6PD through phosphorylation, and thereby G6PD's dimerization, Plk-1 directly affects glucose flux through the PPP and thus the rate of biosynthesis mediated by ribose-5-P production. The results provide very useful mechanistic insight into an important potential drug target that might impact a range of cancers. The results are impressive and appear to make a compelling story. To this reviewer, the experiments are well done and on the whole reasonably well described.

One question I had was why the Plk1 knockdown experiments (Fig 1n) decreased the lactate production from PPP by about 1/3. I might have expected that the effect would be somewhat larger. Maybe there is another regulator of G6PD?

I would also appreciate seeing some of the raw NMR and LC-MS data used to generate the various plots in the Figures. These data (such as a 2D NMR HSQC spectrum, chemical shift information, LC retention time, accurate mass m/z, etc.) should be added to the Supplementary Information section. For example, a number of ¹³C labeled nucleotides, as well as ¹³C labeled lactate are described, but no chemical shift assignments are given, and thus it is impossible for an interested reader to verify that proper identification was made, or to make an assessment of the data quality in terms of signal to noise. To this point, how many cells were used in the LC-MS and NMR measurements? This latter information should be provided in the Methods Section.

A related point is that the quantitation of metabolites using HSQC spectra is complicated by the fact that the metabolites can have different relaxation times. How did the authors account for this?

Overall, an impressive study worthy of publication after the above issues are addressed.

Reviewer #3 (Remarks to the Author):

In the manuscript by Ma et. al., the authors identified that PLK1 phosphorylates G6PD and promotes its dimerization, which leads to the upregulation of G6PD activity. PLK1-mediated

G6PD activation is important for glucose flux through the pentose phosphate pathway to sustain nucleotide biosynthesis. Depletion of PLK1 or G6PD inhibits cell cycle progression in vitro and suppresses tumor growth in vivo. The novel of the manuscript resides in the identification of PLK1-mediated G6PD phosphorylation and its regulation on enzyme dimerization. The findings provide additional molecular insight into the coordination between cell cycle and metabolism programs in cancer cells. However, several major concerns remain to be addressed.

1. The authors showed the cell cycle-dependent changes in NADPH level and G6PD activity. However, the authors didn't provide solid evidence that G6PD activity change is the cause for the fluctuation of NADPH level during cell cycle progression. What about the activity of other NADPH generating pathways, such as MTHFD1 and malic enzyme?

2. The authors showed the regulation of NADPH level by cell cycle and PLK1. On the other hand, NAC has minimal rescue effect upon PLK1 depletion. The authors should show whether the change in NADPH level is correlated with changes in GSH/GSSG and cellular ROS level.

3. Based on the description in Fig.1 legend, the cells were changed into C-13 labeling medium one hour before the cells were released from HU block into G1, S or G2/M phase. This means the labeling times for the samples in Fig.2K. Please clarify the experimental details. In addition, R5P is synthesized from glucose through both oxidative and non-oxidative pentose phosphate pathway. The authors should provide the flux data for all major glucose metabolism pathways and glucose uptake data to demonstrate whether the changes in R5P is the consequence of change in G6PD activity, general decrease in glycolysis activity or suppression of glucose uptake. Along similar line, the authors should show additional flux data on glycolysis and pentose phosphate pathway from the 2C13-glucose labeling experiment and measure overall lactate production upon PLK1 knockdown or overexpression.

4. No detailed description of the measurement of mitotic time with live cell imaging was provided in Methods or Figure legends. Is the data in Fig.4a indicative of the duration of M phase? If the major function of G6PD is to provide precursor for nucleotide biosynthesis, the most affected cell cycle should be S phase. Why is G2/M phase mostly affected?

5. PLK1 can affect the progression G2/M phase through its well-known effect on cytokinesis which may explain G2/M arrest upon PLK1 knockdown in Fig.4D. However, it's intriguing why nucleoside could rescue the G2/M arrest and decrease in cell proliferation upon PLK1 knockdown. Does nucleoside also rescue the block of cytokinesis upon PLK1 depletion?

6. The authors should provide the data for cells infected with control shRNA (NTC in Fig.4A) as additional control in Fig.4C.

Point-by-point response to the comments

First of all, we appreciate greatly the constructive comments and suggestions from the reviewers. Accordingly, we have now performed additional new experiments and addressed all the concerns and comments. Our point-by-point responses are appended below.

For the reviewer's convenience, we have appended in this file all the revised figures, which we labeled as **Fig. R1** to **Fig. R11** and **Table R1** to **R3** and **Appendixes RI-RII**.

Reviewers' comments:

Reviewer #1 (Remarks to the Author):

In this study, the authors have investigated the role of polo like kinase 1 (Plk1) in biosynthesis in cancer cells. Plk1 is a serine/threonine kinase and a key regulator of mitosis and cell cycle progression. It is now known that besides cell cycle regulation Plk1 plays other important roles in regulating metabolism and cancer cell metastasis. This study suggests that Plk1 activates pentose phosphate pathway (PPP) in cancer cells by directly phosphorylating glucose-6-phosphate dehydrogenase (G6PD), a rate limiting enzyme of PPP. Activation of G6PD by Plk1 increase PPP flux to produce NADPH and ribose via glucose metabolism. The authors further demonstrate that the activation of G6PD by Plk1 is critical for cell cycle progression and cancer cell growth. The authors have produced results by utilizing both in vitro cancer cells and in vivo xenograft mouse model. The authors have made an attempt to investigate and establish the coordination between cell cycle progression and metabolic reprogramming with a common link of Plk1 in these two cellular pathways important for cancer progression.

Overall, the study is interesting and the manuscript is written and presented well. The results are presented nicely with necessary statistical analysis and discussed well in the manuscript. However, the following concerns need to be addressed by the authors.

Response: We are grateful for the reviewer's comments that well summarized the major findings and significance of our study. Indeed, we discover here an interesting coordination between cell cycle and metabolism that is mediated by Plk1. This discovery is novel and of significance. Meanwhile, we have performed additional experiments to address the reviewer's concerns and comments. We thank the reviewer for helping us to improve the manuscript substantially.

Specific comments:

Comments-1-1

1. Some recent studies also investigated the role of Plk1 in cancer cell metabolism. These studies should be discussed in this manuscript.

Response: We appreciate the reviewer's suggestion to discuss further recent studies that link Plk1 to cancer cell metabolism. In the original manuscript, we've already discussed the important literature by Li Z et al who reported that PTEN phosphorylation by Plk1 leads to its inactivation, and subsequent activation of the PI3K pathway and enhanced aerobic glycolysis (Li Z, et al., 2014). Following the reviewer's suggestion, in the Introduction section of the revised manuscript, we cited another recently published paper reporting knockdown of Plk1 alters metabolic regulation in melanoma (Gutteridge, R. E., et al., 2017). Moreover, in the Discussion section of the revised manuscript, we further discussed the report by Shao C et al that BI2536 treatment inhibits metformin-induced glycolysis and glutamine anaplerosis. (Shao C, et al., 2015). Nevertheless, it's worthwhile to note that, while those studies linked Plk1 to metabolic regulation in cancer cells, our study demonstrate for the first time that Plk1 coordinates biosynthesis during cell cycle progression by directly activating G6PD and pentose phosphate pathway.

Comments-1-2

2. Authors have used Plk1 inhibitor BI2536 in in vitro studies, however, they have not used this inhibitor for in vivo studies. It would be interesting to see if BI2536 has any effect on G6PD in xenograft model.

Response: Following this suggestion, we performed additional experiments to examine the effect of Plk1 inhibitor BI2536 on G6PD *in vivo* using xenograft model. Consistent with the data in the original manuscript, overexpression of G6PD promoted tumor growth *in vivo*, which were abolished by BI2536 treatment (**Fig. R1a**). Using the tumor tissues, we further confirmed that BI2536 treatment suppressed G6PD activity *in vivo* (**Fig. R1b**) while it exerted no effect on G6PD protein levels (**Fig. R1c**). These data further demonstrated that BI2536 suppressed G6PD activity *in vivo* to retard tumor growth. However, it's worthwhile to point out that BI2536 might also suppress tumor growth via mechanisms not related to G6PD enzyme activity since the inhibition was observed in both control and G6PD overexpression groups. We believe this to be no surprise considering the known effects of Plk1 on mitosis and so on. These new data were incorporated into the revised manuscript (**Fig. S5d-5f in the revised manuscript**).

Fig. R1. Plk1 inhibitor BI2536 suppresses G6PD activity and tumor growth *in vivo*. (a-c) Equal numbers of HeLa cells stably expressing empty vector (EV) or G6PD were subcutaneously injected into Balb/c nude mice (n=5 for each group). Vehicle (saline) or BI2536 (35 mg/kg body weight) were injected intraperitoneally one day before cell inoculation and subsequently twice a week. Tumor volumes were measured starting from 8 days after cell inoculation (a). Tumors were extracted followed by G6PD activity measurement (b) and protein analysis (c) at the end of the experiment. *P<0.05 as compared between indicated groups by two-sided student's t-test; values were represented as the mean±s.d. (see also Fig. S5d-5f in the revised manuscript).

Comments-1-3

3. What was the concentration used for Plk1 inhibitor BI2536? This should be given either in fig 1 legend or on the figure itself.

Response: The concentration used for Plk1 inhibitor BI2536 was 1 μM. We have provided this information in the Figure 1 legend and on the Figure 1f itself.

Comments-1-4

4. Figure 1a. Western blot for G6PD is not convincing. In the result section, the authors suggest that there was no variation G6PD protein expression during cell cycle progression. However, G6PD expression seems to be downregulated in blot in G2/M phase compared to G1 and S phase. This downregulation could negatively

affect activity of G6PD which is shown significantly increased in G2/M phase in fig. 1b. Better blot should be provided in this regard, or authors should provide densitometry data of blots normalized with loading control.

Response: To make this clear, we have actually performed more than three biological replicates of HeLa cells to examine the protein levels of G6PD and other PPP enzymes during cell cycle progression. As you may see from the blot data below, there was indeed no significant variation in G6PD protein expression during cell cycle progression (**Fig. R2**). To avoid confusions, we have followed the reviewer's suggestion to replace the figure by a better blot (**blot b**) in the revised manuscript (**Fig. 1a in the revised manuscript**).

Fig. R2. The protein levels of oxidative pentose phosphate pathway enzymes were not changed during cell cycle progression. (a-c) HeLa cells were synchronized to G1, S and G2/M phases. Cells were harvested and subjected to determine protein levels by western blot as indicated. Three independent biological replicates were performed. β -actin served as the loading control.

Comments-1-5

5. Fig 2a. There is a variation in beta-actin blot. It should be replaced by a better blot.

Response: We appreciate the reviewer for pointing this out. We have replaced the beta-actin blot by a better one (**blot c**) since we had performed three biological replicates (**Fig. R3 or Fig. 2a in the revised manuscript**).

Fig. R3. The western blot results of G6PD and Plk1 in different HeLa cells as indicated. (a-c) HeLa cells stably expressing NTC or shG6PD were infected with viruses expressing empty vector (EV) or Plk1. Cells were harvested and subjected to western blot. Three independent biological replicates were performed. β-actin served as the loading control.

Comments-1-6

6. In some figures, the stars (*) on the graphs are not in proper place. Please correct that wherever required such as in fig. 1h, 1k, 2d and suppl 4d.

Response: We apologize for this error that leads to confusion. In the revised figures, the asterisks were positioned in the right place (Please see **Fig. 1h, 1k, 2d and Fig. S4d in the revised manuscript**).

Comments-1-7

7. Suppl fig 1c, scale bar and magnification should be provided on the immunofluorescence images. In the same figure, nucleus stained with DAPI are not clearly visible. Authors may provide high-power images for the same.

Response: We thank the reviewer for picking up this detail. Now the scale bar and magnification were provided and the DAPI images were refined (**Fig. R4 or Fig. S1c in the revised manuscript**).

Fig. R4. Immunofluorescence staining of pH3Ser10 in synchronized HeLa cells treated with BI2536 or vehicle. HeLa cells were synchronized into G1 phase and released to fresh DMEM for 0, 5, 10, 12 hours. BI2536 or DMSO was added 1 hour before releasing. Cells were subjected to immunofluorescence assay to stain mitotic marker pH3Ser10 (see also Fig. S1c in the revised manuscript).

Comments-1-8

8. Suppl figure 2h, the 3rd figure of phosphor-site S296 is very fuzzy and not readable.

Response: Following the reviewer's suggestion, the image of phosphor-site S296 was replaced by a better one with higher resolution (Fig. R5 or Fig. S2h in the revised manuscript).

phospho-site S296

Spectrum from G6FD-2.wiff (sample 1) - Sample003, Experiment 22, *TOF MS*2 (100 - 1600) from 31.319 min
Precursor: 975.1 Da

Fig. R5. LC-MS/MS identified phosphorylation sites of Plk1 on G6PD. The phosphorylation site (S296) of Plk1 on G6PD was identified by mass spectrometry (see also **Fig. S2h** in the revised manuscript).

Comments-1-9

9. Authors should more carefully proofread for grammatical mistakes. Ex. Legend to suppl fig 2(a-b). Hep3B cells “were infection” with virus, should be corrected to “were infected”.

Response: We have carefully proofread the entire manuscript and corrected as much as we could the typo errors and grammatical mistakes in the revised manuscript. Thank you for all your constructive comments and suggestions that helped us to improve the manuscript significantly.

Reviewer #2 (Remarks to the Author):

Ma et al. report on a series of careful experiments designed to investigate the effects of Plk-1 on the pentose phosphate pathway during cell cycle progression. They show that in its role in activating G6PD through phosphorylation, and thereby G6PD's dimerization, Plk-1 directly affects glucose flux through the PPP and thus the rate of biosynthesis mediated by ribose-5-P production. The results provide very useful mechanistic insight into an important potential drug target that might impact a range of cancers. The results are impressive and appear to make a compelling story. To this reviewer, the experiments are well done and on the whole reasonably well described.

Response: We appreciate the reviewer for the positive comments pointing out the significance and novelty of our study.

Comments-2-1

One question I had was why the Plk1 knockdown experiments (Fig 1n) decreased the lactate production from PPP by about 1/3. I might have expected that the effect would be somewhat larger. Maybe there is another regulator of G6PD?

Response: We thank the reviewer for raising this relevant question. We understand that the reviewer might expect a larger decrease in the level of lactate production following Plk1 knockdown. However, considering that lactate is the final product of glycolysis, the decrease of about 1/3 in its production by Plk1 knockdown is really significant. Moreover, the shRNA-mediated knockdown can't deplete the gene expression of Plk1 completely, thus potentially limiting its effects on G6PD and

downstream functions. Indeed, as the reviewer has pointed out, there exist other molecules such as P53 in regulation of G6PD and PPP pathway (Already discussed in the manuscript). Nevertheless, based on our multiple lines of evidence, Plk1 markedly regulates G6PD activity by interacting with and phosphorylating G6PD.

Comments-2-2

I would also appreciate seeing some of the raw NMR and LC-MS data used to generate the various plots in the Figures. These data (such as a 2D NMR HSQC spectrum, chemical shift information, LC retention time, accurate mass m/z, etc.) should be added to the Supplementary Information section. For example, a number of ¹³C labeled nucleotides, as well as ¹³C labeled lactate are described, but no chemical shift assignments are given, and thus it is impossible for an interested reader to verify that proper identification was made, or to make an assessment of the data quality in terms of signal to noise. To this point, how many cells were used in the LC-MS and NMR measurements? This latter information should be provided in the Methods Section.

Response: Per the request of the reviewer, we have provided the NMR and LC-MS raw data as well as the qualitative and quantitative methods (**Fig. R6, Table R1-R2 and Appendix RI-RII**). **Table R1 and R2** were also included in the Supplementary Information in the revised manuscript (**Table s1 to s2 in the Supplementary Information**).

For NMR data: We provided the metabolite assignments of the 2D HSQC spectra in **Fig. R6a and 6b**. Detailed chemical shift information of individual metabolite can be found in **Table R1**. For space consideration, all assignment-related NMR spectra and HSQC spectra of standards and cell extracts and the raw data of the peak volumes (normalized by cell wet weight) of the cross-peaks in 2D HSQC spectra are provided in the **Appendix RI**.

For LC-MS data: We provided retention time and accurate m/z information of each metabolite in **Table R2**. For space consideration, the raw LC-MS data as well as qualitative and quantitative methods were provided in the **Appendix RII**.

The cells we used: For NMR measurements, the peak volumes of the cross peaks were normalized to cell wet weights. The cell numbers are estimated to be roughly 3×10^7 per 100 mg wet weight. For most samples, 80-100 mg of wet weight cells (roughly $2.5-3 \times 10^7$ cells/sample) were used. For LC-MS, the cell number we used is 5×10^6 . We have provided this cell number information in the Method Section in the revised manuscript.

Fig. R6. The HSQC spectra and assignments (a) HSQC spectrum of 2-¹³C glucose labeled sample (b) HSQC spectrum of U-¹³C glucose labeled sample.

Table R1: Chemical shift of each metabolite

Compound	Group	Structure	$\delta(^1\text{H})/\text{ppm}$	$\delta(^{13}\text{C})/\text{ppm}$
R5P	C1		5.24	104.08
CXP	C1		6.15	99.47
UXP	C1		5.98	105.49
GXP	C1'		5.95	89.53
AXP	C1'		6.14	89.79
Lactate	C3		1.32	23.08
	C2		4.10	72.08
G6P	C2		3.27	76.94
Pyruvate	C3		2.35	29.80

See also Table s1 in the revised manuscript.

Table R2: Retention time and accurate m/z of each metabolite

Metabolite	RT	M/Z
R5P	5.701	299.0119 +/- 0.0025Da
AMP	5.046	346.056 +/- 0.010Da
dTMP	4.040	321.049 +/- 0.010Da
dCMP	5.517	306.050 +/- 0.005Da

See also Table s2 in the revised manuscript.

Comments-2-3

A related point is that the quantitation of metabolites using HSQC spectra is complicated by the fact that the metabolites can have different relaxation times. How did the authors account for this?

Overall, an impressive study worthy of publication after the above issues are addressed.

Response: As the reviewer pointed out, 2D cross-peak intensities (or volumes) are influenced by a greater number of variables (e.g., uneven excitation, non-uniform relaxation, evolution times, mixing times, etc.), which makes it difficult to translate peak intensities into absolute metabolite concentrations. Lewis et al proposed a FMQ approach (FMQ: fast metabolite quantification). For each pre-identified target metabolite resonance, the non-uniform behavior including relaxation effects for cross-peaks of individual metabolites are expected to be the same, considering the fact that all cell extracts were under the same treatment and the same NMR acquisition parameter set were applied (Lewis, et al., 2007). According to this FMQ method, a standard curve can be constructed for each metabolite by regressing absolute peak intensities from the concentration reference samples with their known concentrations, and accurate (technical error 2.7%) molar concentrations can be determined. In this study, we followed the general guideline of FMQ method, except that we are not trying to determine the absolute metabolite concentration. Instead, we are interested in the relative concentration changes of the same metabolite in various cell samples. Therefore, the good linear correlation between resonance intensity and corresponding concentration that was demonstrated in Lewis paper, set a solid foundation for characterizing the metabolite concentration changes by their relative intensity changes in 2D NMR signals. Here we utilize the changes in 2D cross-peak intensities to quantify their concentration changes. For metabolites like ATP, ADP and AMP, some of the resonances are identical, which may bring in quantitation complexity if they have different relaxation times. In this study, we carefully measured the T1 relaxation times of AXPs, CXPs, UXPs and GXPs, and assessed the potential errors associated with relaxation delay. For detailed information please see Appendix RII and Table R3 (Table s3 in the revised Supplementary Information).

Table R3: T1 experiments and calculation

298k, 10mM	ATP	ADP	AMP	CTP	CDP	CMP	GTP	GDP	GMP	UTP	UDP	UMP
T1	8.70	7.85	8.79	7.97	6.88	6.86	5.69	7.30	7.65	7.29	7.93	8.57
M(t)/M(0), * t=1.5s	0.16	0.17	0.16	0.17	0.20	0.20	0.23	0.19	0.18	0.19	0.17	0.16
SD		0.01			0.01			0.03			0.01	

* M(t) and M(0) are the magnetization at the time t and equilibrium state, t: relaxation delay. See also Table s3 in the revised manuscript.

Reviewer #3 (Remarks to the Author):

In the manuscript by Ma et. al., the authors identified that PLK1 phosphorylates G6PD and promotes its dimerization, which leads to the upregulation of G6PD activity. PLK1-mediated G6PD activation is important for glucose flux through the pentose phosphate pathway to sustain nucleotide biosynthesis. Depletion of PLK1 or G6PD inhibits cell cycle progression in vitro and suppresses tumor growth in vivo. The novel of the manuscript resides in the identification of PLK1-mediated G6PD phosphorylation and its regulation on enzyme dimerization. The findings provide additional molecular insight into the coordination between cell cycle and metabolism programs in cancer cells. However, several major concerns remain to be addressed.

Response: We appreciate the reviewer for pointing out the novelty and significance of this manuscript. Indeed, our study provides additional molecular insight into the coordination between cell cycle and metabolism in cancer cells. Meanwhile, we are also grateful for his/her multiple critical comments which have helped us to substantially improve the manuscript. We have addressed all the comments/concerns below and revised the manuscript.

Comments-3-1

1. The authors showed the cell cycle-dependent changes in NADPH level and G6PD activity. However, the authors didn't provide solid evidence that G6PD activity change is the cause for the fluctuation of NADPH level during cell cycle progression. What about the activity of other NADPH generating pathways, such as MTHFD1 and malic enzyme?

Response: Indeed, there exist multiple NADPH generating pathways, which involve many other metabolic enzymes such as MTHFD1, ME (malic enzyme) and IDH (Isocitrate dehydrogenase). To address the reviewer's questions experimentally, we first measured the protein levels of these NADPH generating enzymes in synchronized cells or in cells with Plk1 overexpression/knockdown, and detected no significant variations in protein levels of MTHFD1, ME1 or IDH2 (**Fig. R7a-c**). Next, we measured the activity of ME and IDH, as a result, no significant activity changes were observed for these enzymes in synchronized cells or in cells with different Plk1 expression (**Fig. R7d-i**). These results, together with our observation that G6PD activity fluctuates during cell cycle progression or with Plk1 protein expression, further suggest that G6PD is likely the enzyme whose activity changes represent the cause for the fluctuation of NADPH level during cell cycle progression. Thus, we further synchronized HeLa cells expressing NTC or shG6PD and measured NADPH

levels. The results showed that the increment of NADPH during cell cycle progression was significantly blocked by shG6PD (Fig. R7j), further confirming that G6PD is important for the fluctuation of NADPH during cell cycle progression. Of note, because no commercial kit is currently available to us, unfortunately, we were unable to measure the activity of MTHFD1.

Fig. R7. The protein levels and activities of other NADPH generating enzymes during cell cycle progression. (a-c) Protein levels of the major NADPH generating enzymes were determined by western blot in HeLa cells synchronized into G1, S and G2/M phases (a), or in HeLa cells overexpressing Plk1 (b) or expressing tet-inducible shPlk1 (c). (d-i) Enzyme activities of malic enzyme (ME) and Isocitrate dehydrogenase (IDH) were measured in HeLa cells synchronized into G1, S and G2/M phases (d-e), or in HeLa cells overexpressing Plk1 (f-g), or expressing tet-inducible shPlk1 (h-i). (j) HeLa cells expressing NTC or shG6PD were synchronized into G1, S and G2/M phases. Cells were harvested and NADPH levels were measured by commercial kit. * $P < 0.05$ as compared to G1-NTC group, # $P < 0.05$ as compared to corresponding NTC group, by two-sided student's t-test. ns: not significant. Values represent the mean \pm s.d. β -actin serves as the loading control.

Comments-3-2

2. The authors showed the regulation of NADPH level by cell cycle and PLK1. On the other hand, NAC has minimal rescue effect upon PLK1 depletion. The authors should show whether the change in NADPH level is correlated with changes in GSH/GSSG and cellular ROS level.

Response: As suggested by the reviewer, we performed additional assays to measure GSH and ROS levels in HeLa cells with Plk1 overexpression or knockdown. The results showed that, consistent with the changes of NADPH, Plk1 expression affected the levels of GSH and ROS as well (**Fig. R8 or Fig. S1g-1k in the revised manuscript**), indicating that, by regulating G6PD-mediated biomass synthesis, Plk1 had an effect on cellular redox status via NADPH/GSH/ROS. However, the role of Plk1 on PPP pathway and biosynthesis is obviously not limited to NADPH. Thus, while we did observe that NAC exerted some rescue effects upon Plk1 depletion, the

effects of nucleotides were more obvious in our experimental systems. NAC alone was not very efficient. We actually did all subsequent rescue experiments with Nuc and NAC mixture (Fig. 4 and Fig. S4 in the original manuscript).

Fig. R8. Plk1/G6PD regulates cellular GSH and ROS levels (a-c) GSH levels were measured in HeLa cells overexpressing Plk1 (a), or expressing shPlk1 (b) or in Plk1 overexpressing HeLa cells with further G6PD knockdown (c). (d-e) ROS levels were measured in HeLa cells expressing tet-inducible shPlk1 (d) or in HeLa cells overexpressing Plk1 (e). *P<0.05 as compared to EV or NTC group by two-sided student's t-test; values represent the mean±s.d. See also Fig. S1g-1k in the revised manuscript)

Comments-3-3

3. Based on the description in Fig.1 legend, the cells were changed into C-13 labeling medium one hour before the cells were released from HU block into G1, S or G2/M phase. This means the labeling times for the samples in Fig.2K. Please clarify the experimental details. In addition, R5P is synthesized from glucose through both oxidative and non-oxidative pentose phosphate pathway. The authors should provide the flux data for all major glucose metabolism pathways and glucose uptake data to demonstrate whether the changes in R5P is the consequence of change in G6PD

activity, general decrease in glycolysis activity or suppression of glucose uptake. Along similar line, the authors should show on glycolysis and pentose phosphate pathway from the 2C13-glucose labeling experiment and measure overall lactate production upon PLK1 knockdown or overexpression.

Response: The reviewer is right about the labeling time of the samples in Fig. 1k. Following the reviewer's suggestion, we have provided more experimental details in the legend of Fig. 1k in the revised manuscript. Briefly, HeLa cells were synchronized by double HU block which was described in Methods section of the manuscript. As shown in **Fig. R9a**, when the cells were treated for the second round of HU block for 11 hours, i.e. 1 hour before releasing, U-¹³C glucose was added into the cell culture medium and further culture for 1 hour and then harvested for G1 phase cell analysis. For cell samples of S and G2/M phases, after the regular second round of HU block for 12 hours, cells were released into fresh HU-free medium and further cultured for 4 hours and 9 hours, respectively, followed by supplementation of U-¹³C glucose and 1 hour culture before harvesting for S and G2/M phases cell analysis.

As the reviewer has pointed out, R5P is synthesized from glucose through both oxidative and non-oxidative pentose phosphate pathway. Our data indicate that Plk1 expression may affect the production of R5P from ¹³C-labeled glucose, however, these data could not distinguish whether the production of R5P is from the oxidative or non-oxidative pentose phosphate pathway. This is very complicated. As shown in **Fig. R9b**, there are multiple combinations of R5P generation when tracing the carbon using 2-¹³C glucose via general glycolysis or pentose phosphate pathways. While the oxidative PPP is simple and more straightforward, the non-oxidative one could generate many potential products, whose signal via ¹³C-labeling tracing could be very weak. Moreover, both the oxidative and non-oxidative PPP could potentially generate the same 1-¹³C-R5P product from 2-¹³C-glucose. As a result, it is very difficult for us to distinguish R5P from the two pathways even using 2-¹³C-glucose labeling experiments.

However, following the reviewer's comments, we performed additional experiments to further clarify this important point. We knocked down G6PD in HeLa cells overexpressing Plk1 to suppress the oxidative PPP activity but not the non-oxidative one, and performed U-¹³C glucose metabolic flux assay. As a result, Plk1 overexpression significantly enhanced R5P production, which was abolished by shG6PD (**Fig. R9c**). Consistently, shG6PD also significantly abolished the Plk1 overexpression-induced increase in the production of GXP, UXP, CXP and AXP (**Fig. R9c**). Collectively, these data demonstrate that G6PD-mediated oxidative pentose phosphate pathway was the major cause for changes in R5P and subsequent

nucleotides levels regulated by Plk1. We have included these new data in the revised manuscript (**Fig. 1q in the revised manuscript**).

C
Fig. R9. Plk1 promotes oxidative pentose phosphate pathway. (a) The time table for preparing G1, S and G2/M synchronized cell samples in U-¹³C glucose labeling experiment. (b) The ¹³C carbon flow of 2-¹³C glucose through glycolysis and PPP. (c) U-¹³C glucose metabolic flux assays were performed in Plk1 overexpressing HeLa cells with G6PD knockdown by NMR. *P<0.05 as compared to corresponding EV-NTC group by two-sided student's t-test; values were represented as the mean± s.e.m. (See also **Fig. 1q** in the revised manuscript).

Following the reviewer's suggestion, we also measured changes in glucose uptake as well as the levels of some glycolytic metabolites in HeLa cells and found that Plk1 overexpression enhanced glucose uptake and the production of lactate, G6P and pyruvate, indicating that Plk1 did have effects on glycolysis in general (**Figure R10a-10f**). This is consistent with a previous report by Li Z et al, showing that Plk1 regulates glycolysis in cancer cells via targeting PTEN/PI3K pathway (Li et al., 2014). However, our study showed clearly that Plk1 activates G6PD and oxidative pentose phosphate pathway to promote biosynthesis and cancer progression. We have included these new data in the revised manuscript (**Fig. S10-1t** in the revised manuscript).

Fig. R10. Plk1 regulates glucose uptake and glycolysis. (a-d) Glucose uptake or overall lactate production was measured in HeLa cells overexpressing Plk1 (a, c) or expressing tet-inducible shPlk1 (b, d). (e-f) ^{13}C -incorporated G6P and pyruvate were measured in $2\text{-}^{13}\text{C}$ glucose metabolic flux assay by NMR in HeLa cells overexpressing Plk1 (e) or expressing tet-inducible shPlk1 (f). * $P < 0.05$ as compared to corresponding EV or -tet group by two-sided student's t-test; values were represented as the mean \pm s.d. or s.e.m. (See also Fig. S10-1t in the revised manuscript).

Comments-3-4

4. No detailed description of the measurement of mitotic time with live cell imaging

was provided in Methods or Figure legends. Is the data in Fig.4a indicative of the duration of M phase? If the major function of G6PD is to provide precursor for nucleotide biosynthesis, the most affected cell cycle should be S phase. Why is G2/M phase mostly affected?

Response: We thank the reviewer for pointing this out. We have now provided detailed description of the measurement of mitotic time with live cell imaging in Method section and figure legends in revised manuscript. Yes, the data in **Fig. 4a** are indicative of the duration of M phase.

As a matter of fact, our results showed that both S phase and G2/M phases were affected by Plk1 and that addition of Nuc and NAC could significantly rescue both the S phase and G2/M phase cell population suppressed by shPlk1 (**Fig. 4d**). Indeed, we confirmed repeatedly that G6PD activity starts to increase during S phase, but reaches the peak level at G2/M instead of S phase, we don't know why exactly. It's likely that, during cell cycle progression, the biosynthesis process continues beyond S phase, as Atilla-Gokcumen documented recently that the lipid composition of cells in S phase differs drastically from cells in cytokinesis (Atilla-Gokcumen, et al., 2014). We believe the reviewer raised here a very stimulating question, therein potentially lies paradigm-shifting mechanisms and answers for the control of cell cycle progression.

Comments-3-5

5. PLK1 can affect the progression G2/M phase through its well-known effect on cytokinesis which may explain G2/M arrest upon PLK1 knockdown in Fig.4D. However, it's intriguing why nucleoside could rescue the G2/M arrest and decrease in cell proliferation upon PLK1 knockdown. Does nucleoside also rescue the block of cytokinesis upon PLK1 depletion?

Response: Again, this is a very stimulating question. At this moment, we don't have any evidence demonstrating that nucleosides also rescue the block of cytokinesis upon Plk1 depletion. However, it's reasonable for us to believe that, when the knockdown effect of Plk1 is not complete with shPlk1 and so is its blocking effect on cytokinesis, the addition of nucleosides could partially rescue the S and G2/M arrest and cell proliferation, as we have observed in **Fig. 4d-4e**.

Comments-3-6

6. The authors should provide the data for cells infected with control shRNA (NTC in Fig.4A) as additional control in Fig.4C.

Response: We thank the reviewer for the suggestion. Accordingly, we performed new experiments to include the NTC-EV group as an additional control. The results showed that the cell number of NTC-EV group was similar to that of the wild-type group (Fig. R10). We also replaced the Fig. 4c in the revised manuscript.

Fig. R11. The effects of G6PD mutation on cell proliferation. HeLa cells stably expressing NTC or shG6PD were further infected with viruses expressing empty vector (EV) or Flag-G6PD wild-type or its mutants as indicated. Cell were treated with vehicle or Nuc mix and NAC supplementation. Cell numbers were counted 4 days after treatment. * $P < 0.05$ as compared between indicated groups by two-sided student's t-test; values were represented as the mean \pm s.d. (See also Fig. 4c in the revised manuscript).

References:

1. Li Z, *et al.* Plk1 Phosphorylation of PTEN Causes a Tumor-Promoting Metabolic State. *Molecular and cellular biology* 34, 3642-3661 (2014).
2. Shao C, Ahmad N, Hodges K, Kuang S, Ratliff T, Liu X. Inhibition of polo-like kinase 1 (Plk1) enhances the antineoplastic activity of metformin in prostate cancer. *Journal of Biological Chemistry* 290, 2024-2033 (2015).
3. Gutteridge REA, Singh CK, Ndiaye MA, Ahmad N. Targeted knockdown of polo-like kinase 1 alters metabolic regulation in melanoma. *Cancer letters* 394, 13-21 (2017).
4. Lewis, I. A., *et al.* Method for determining molar concentrations of metabolites in complex solutions from two-dimensional H-1-C-13 NMR spectra. *Analytical Chemistry* 79(24): 9385-9390 (2007).
5. Atilla-Gokcumen GE, *et al.* Dividing cells regulate their lipid composition and localization. *Cell* 156, 428-439 (2014).

Appendix RI:

NMR HSQC spectra and quantification of each metabolite.

The HSQC spectra of each metabolite standard.

R5P:

AXP:

GXP:

CXP:

UXP:

G6P:

Lactate:

Pyruvate:

T1 measurement using inversion recovery:

* $M(t)$ and $M(0)$ is the magnetization at the time t and equilibrium state, RD: relaxation delay.

The HSQC spectrum of 2-¹³C labeled sample:

The HSQC spectrum of U-¹³C labeled sample:

The normalized integrals of our raw NMR data:

For 2-¹³C glucose labeled samples (Fig. 1m-1n, Fig. S1s-1t):

the integrals of each metabolite				integrals			
No.	sample	labeling	replicates	G6P	Pyruvate	Lactate 2-C	Lactate 3-C
1	EV	2- ¹³ C glucose	1	1675580.1	6109502.76	38381215.47	44082872.93
2	EV	2- ¹³ C glucose	2	2168379.8	3239198.61	43921602.79	51533101.05
3	EV	2- ¹³ C glucose	3	2584292.6	11970263.8	95436450.84	149280575.5
4	Plk1	2- ¹³ C glucose	1	3333333.3	6736178.86	57593495.93	59086720.87
5	Plk1	2- ¹³ C glucose	2	2861279.8	5103036.88	47207158.35	48849240.78
6	Plk1	2- ¹³ C glucose	3	2703689.6	16821883	60035623.41	99950381.68
7	tet-	2- ¹³ C glucose	1	6763681.6	19063018.2	74338498.21	141358760.4
8	tet-	2- ¹³ C glucose	2	10019854	12159380.7	7301854.305	11351258.28
9	tet-	2- ¹³ C glucose	3	6121924.5	7712302.07	25910569.11	38889227.64
10	tet+	2- ¹³ C glucose	1	4073897.5	14576877.2	136867330	167114427.9
11	tet+	2- ¹³ C glucose	2	7301854.3	11351258.3	31438979.96	81295081.97
12	tet+	2- ¹³ C glucose	3	5595528.5	6629369.92	45308160.78	60544457.98

For U-¹³C glucose labeled samples (Fig. 1o-1q):

No.	sample	labeling	replicates	integrals of each metabolite							
				R5P	CXP	UXP	GXP	AXP	G6P- β 1	G6P- β 2	Pyruvate
1	EV	U-13C glucose	1	8223307.5	5644197.29	143626692.5	20533849.13	45862669.2	38668936.2	17317446.81	96612765.96
2	EV	U-13C glucose	2	6185617	5953446.81	163089361.7	20935319.15	41066383	44183752.4	23572533.85	75792069.63
3	EV	U-13C glucose	3	4913659.5	11304955.5	92401524.78	26693773.82	31661372.3	19209021.6	4359656.925	101581956.8
4	Plk1	U-13C glucose	1	11207692	10237252.7	205560439.6	40848351.65	62017582.4	43098540.1	18231751.82	97198905.11
5	Plk1	U-13C glucose	2	13539234	9002098.54	181861313.9	32114963.5	48680656.9	103674725	59383516.48	106437362.6
6	Plk1	U-13C glucose	3	9159187.1	14637595.3	108704487.7	42217612.19	45245554.6	30329381.9	13461473.33	123700254
7	tet-	U-13C glucose	1	15612590	8294268.17	48984646.88	14694984.65	4491044.01	1234237462	845598771.8	1590122825
8	tet-	U-13C glucose	2	6213359.3	12064461.7	227872892.3	40394293.13	6659662.78	145633397	90307101.73	116619961.6
9	tet-	U-13C glucose	3	7137842.3	14117842.3	243784232.4	33723651.45	70627385.9	46794190.9	33002489.63	46794190.87
10	tet+	U-13C glucose	1	7661503.2	3241250.72	31148594.38	8989099.254	2294205.39	393247275	266815834.8	194394721.7
11	tet+	U-13C glucose	2	3550479.8	4764491.36	160946257.2	19781190.02	3690403.07	86603112.8	57966277.56	55186770.43
12	tet+	U-13C glucose	3	3242939.5	7649951.97	190518732	26819404.42	33437079.7	130297791	91635926.99	100355427.5
13	ntc+ev	U-13C glucose	1	19261084	26502463.1	175233990.1	45570197.04	75123152.7	68772167.5	73658867	98656403.94
14	ntc+ev	U-13C glucose	2	92721228	23960358.1	180895140.7	40195652.17	101088235	159386189	153081841.4	259872122.8
15	ntc+ev	U-13C glucose	3	46240621	19987063.4	185226390.7	39827943.08	59439844.8	94873221.2	86853816.3	131578266.5
16	ntc+ev	U-13C glucose	4	60411465	5969808.92	142305732.5	17347770.7	46182165.6	80892993.6	69231847.13	133222929.9
17	ntc+Plk1	U-13C glucose	1	21091292	37331460.7	211207865.2	56026685.39	180042135	87054775.3	85759831.46	135529494.4
18	ntc+Plk1	U-13C glucose	2	92532951	11394555.9	155673352.4	32522922.64	56932664.8	88866762.2	93767908.31	151762177.7
19	ntc+Plk1	U-13C glucose	3	122267705	143286119	13947167.14	33443342.78	71558073.7	62143059.5	65349858.36	202733711
20	ntc+Plk1	U-13C glucose	4	81168375	10680380.7	156881405.6	31054172.77	57619326.5	99636896	99232796.49	139338213.8
21	shG6PD+EV	U-13C glucose	1	12853012	25572289.2	54307228.92	25920481.93	58492771.1	26569879.5	69086746.99	77034939.76
22	shG6PD+EV	U-13C glucose	2	40897686	14158343.5	45129110.84	17028014.62	37940316.7	33892813.6	84560292.33	66624847.75
23	shG6PD+EV	U-13C glucose	3	31327607	9016073.62	42570552.15	13679754.6	5064090.8	30694478.5	79769325.15	64165644.17
24	shG6PD+EV	U-13C glucose	4	103415274	17373508.4	33091885.44	12492840.1	48998806.7	28368735.1	80659904.53	65813842.48
25	shG6PD+Plk1	U-13C glucose	1	27300000	44408974.4	184038461.5	219217948.7	1129666667	72993589.7	63701282.05	82105128.21
26	shG6PD+Plk1	U-13C glucose	2	74891753	14426546.4	120612113.4	29201030.93	49344072.2	121016753	105319587.6	82051546.39
27	shG6PD+Plk1	U-13C glucose	3	43288265	16127551	113852040.8	31028061.22	53627551	109724490	114506377.6	82875000
28	shG6PD+Plk1	U-13C glucose	4	35593264	13257772	110152849.7	27707253.89	87084196.9	210194301	202396373.1	148354922.3

Appendix RII:

LC-MS raw data and processing:

For metabolites identification via LC-MS, we first run standards for R5P, AMP, dTMP and dCMP to determine the retention time and m/z. The spectra are as below:

R5P-STD

AMP-STD

dTMP-STD

dCMP-STD

Identification information was summarized in the table below.

Metabolite	RT	M/Z
R5P	5.701	299.0119 +/- 0.0025Da
AMP	5.046	346.056 +/- 0.010Da
dTMP	4.040	321.049 +/- 0.010Da
dCMP	5.517	306.050 +/- 0.005Da

For ¹³C-labeling experiments, we take R5P for example to explain how the data was acquired and processed. For R5P, the isotope peak measured was m0, m1 and m5 (the missing isotope peak was counted as 0) (Fig. 1k in the manuscript). The spectra are as follows:

R5P (m0)

R5P (m1)

R5P (m5)

We performed relative quantification using peak area and used ISOCOR software for correcting naturally-occurring isotopes to get the isotopologue distribution. Raw data and processed data are as below:

Peak area

(1h)	G1			S			G2M		
R5P	1	2	3	1	2	3	1	2	3
M+0	11969.83	10965.66	11274.89	21861.72	20281.73	22341.63	18209.7	16586.66	16024.2
M+1	18317.9	16260.48	17936.14	26931.65	29045.73	28185.63	31154.6	30104.82	25310.13
M+2	0	0	0	0	0	0	0	0	0
M+3	0	0	0	0	0	0	0	0	0
M+4	0	0	0	0	0	0	0	0	0
M+5	1890.68	1811	1849.59	4358.67	4160.97	4672.49	5137.87	5137.49	5307.36

Isotopologue distribution

(1h)	G1			S			G2M		
R5P	1	2	3	1	2	3	1	2	3
M+0	0.37742	0.38346	0.36791	0.41937	0.38521	0.41241	0.33764	0.32281	0.34779
M+1	0.56248	0.55273	0.57121	0.49656	0.53516	0.50083	0.56607	0.576	0.53592
M+2	0	0	0	0	0	0	0	0	0
M+3	0	0	0	0	0	0	0	0	0
M+4	0	0	0	0	0	0	0	0	0
M+5	0.0601	0.06382	0.06088	0.08407	0.07963	0.08676	0.0963	0.10119	0.1163

We converted the M+5 value to percentage and the averages of the percentage values of three biological replicates were used in the bar-graph.

R5P				%	%	%		AVE	STDEV
	0.0601	0.06382	0.06088	6.01	6.382	6.088	G1	6.16	0.196173
	0.08407	0.07963	0.08676	8.407	7.963	8.676	S	8.34867	0.360062
	0.0963	0.10119	0.1163	9.63	10.119	11.63	G2/M	10.4597	1.042612

LC-MS spectra and quantification of AMP, dTMP, dCMP. (Fig. S1n in the revised manuscript).

AMP (M0)

AMP (m2)

AMP (m3)

AMP (m4)

AMP (m5)

dTMP (m0)

dTMP (m1)

dTMP (m4)

dTMP (m5)

dCMP (m0)

dCMP (m1)

dCMP (m2)

dCMP (m3)

dCMP (m4)

dCMP (m5)

We also find that, in the original Fig. S1i, we've made a mistake---used the percentage values of one column of individual samples instead of the average value of the three replicates in generating the bar-graph (yellow shade). Now it is corrected in the revised version (green shade) and the trends of these metabolite changes remain the same (Fig. S1n in the revise manuscript).

Data used for bar graph were shown in the excel table below:

replicate	1	2	3		1	2	3		AVE	STDEV
AMP 2h					%	%	%			
	0.15228	0.1533	0.15352		15.228	15.33	15.352	G1	15.3033	0.066161
	0.19213	0.17479	0.18169		19.213	17.479	18.169	S	18.287	0.873002
	0.21509	0.20448	0.23317		21.509	20.448	23.317	G2/M	21.758	1.450617
dTMP 2h					%	%	%		AVE	STDEV
	0.0665	0.070965	0.07543		6.65	7.0965	7.543	G1	7.0965	0.4465
	0.12099	0.115185	0.10938		12.099	11.5185	10.938	S	11.5185	0.5805
	0.1092	0.124435	0.13967		10.92	12.4435	13.967	G2/M	12.4435	1.5235
dCMP 2h					%	%	%		AVE	STDEV
	0.09444	0.095	0.08294		9.444	9.5	8.294	G1	9.07933	0.680695
	0.09345	0.10254	0.11444		9.345	10.254	11.444	S	10.3477	1.05263
	0.10576	0.10998	0.11862		10.576	10.998	11.862	G2/M	11.1453	0.655537
		Former						Corrected		
		AMP	dTMP	dCMP				AMP	dTMP	dCMP
	G1	15.352	7.543	8.294		G1		15.30333	7.0965	9.079333
	S	18.169	10.938	11.444		S		18.287	11.5185	10.34767
	G2/M	23.317	13.967	11.862		G2/M		21.758	12.4435	11.14533
	STDEV	0.066161	0.4465	0.680695		STDEV		0.066161	0.4465	0.680695
		0.873002	0.5805	1.05263				0.873002	0.5805	1.05263
		1.450617	1.5235	0.655537				1.450617	1.5235	0.655537

The bar graph generated:

REVIEWERS' COMMENTS:

Reviewer #1 (Remarks to the Author):

The authors have adequately addressed reviewers' concerns and have revised their manuscript, accordingly. I do not have any additional concerns.

Reviewer #2 (Remarks to the Author):

The authors have answered all of my previous questions and comments.

Reviewer #3 (Remarks to the Author):

The authors have nicely addressed all my concerns and I have no further comment.